# Phytoplankton blooming mechanisms over the East China Sea during post-El Niño summers

Dong-Geon Lee[1,2], Ji-Hoon Oh[2], Jonghun Kam[1,] Jong-Seong Kug*[2,3]

[1]Division of Environmental Science and Engineering, Pohang University of Science and Technology (POSTECH), Pohang, South Korea
[2]School of Earth and Environmental Sciences, Seoul National University, Seoul, South Korea
[3]Interdisciplinary Program in Artificial Intelligence, Seoul National University, Seoul, South Korea

*Correspondence to*: Jong-Seong Kug (jskug@snu.ac.kr)

**Abstract.** During post-El Niño summers, the East China Sea (ECS) has experienced anomalous phytoplankton blooming, but the understanding of associated generating mechanisms remains limited. Here, we analyzed observational (25 years) and long-term simulation data (1,000 years) to investigate the underlying mechanisms for the anomalous phytoplankton blooming in ECS. Results highlight three mechanisms associated with enhanced phytoplankton blooming in ECS during post-El Niño summers: inland runoff, nutrient inputs from the Taiwan Strait (TS), and oceanic sub-surface upwelling-driven blooming mechanisms. Firstly, increased river discharge from the Yangtze River (YR) induces phytoplankton blooms. Secondly, enhanced nutrient inputs through TS promote phytoplankton growth. Additionally, wind-driven Ekman upwelling in ECS provides nutrients for phytoplankton growth from the sub-surface to the surface water layer. These three mechanisms are all linked to the Western North Pacific Anti-Cyclone (WNPAC), suggesting the WNPAC can be a key precursor to predict ECS blooming. Climate model simulations support these complex mechanisms, and thus our results suggest that all mechanisms can contribute to the phytoplankton bloom concurrently.

## 1. Introduction

Numerous rivers, including the Yangtze River (YR), run into the East China Sea (ECS). YR is the longest river in Eurasia, contributing to a shallow continental shelf with high marine primary productivity (Zhai et al., 2023; Zhang, 1996; Liu et al., 2003; Zhao and Guo, 2011; Tong et al., 2015; Liu et al., 2010). Particularly, the Yellow Sea (YS) and ECS, which belong to the East Asian Marginal Seas (EAMS), are known for being among the most productive marine environments globally. Nutrients for phytoplankton growth are ample from the intrusion of nutrient-rich Kuroshio intermediate water through the Taiwan Strait (TS; Chen, 1996; Chen et al., 1995; Zhang et al., 2007), atmospheric deposition (Kim et al., 2011; Zhang et al., 2010), and primarily riverine input from the YR (Wang et al., 2003; Huang et al., 2019).

Marine phytoplankton plays a role in the marine trophic chain (Danielsdottir et al., 2007), ocean carbon, and biogeochemical cycles (Field et al., 1998; Behrenfeld et al., 2006). Chlorophyll-a (Chl-a) concentration is widely used as a proxy for phytoplankton biomass and it has strength for handiness to measure from satellite (Henson et al., 2010). The ECS region is surrounded by highly populated and developed nations with hundreds of millions of people, which can have a profound impact on fisheries and marine ecosystems. Therefore, understanding the spatio-temporal variability of phytoplankton in the ECS region has great socio-economic importance, particularly in coastal communities.

ENSO plays a key role as a climate regulator for East Asia through its teleconnection. The Western North Pacific AntiCyclone (WNPAC) regulates the East Asian climate during ENSO events from the developing to decaying phases (Wang et al., 2000; Son et al., 2014; Li et al., 2017; Kim and Kug, 2018). ENSO-induced changes in the atmospheric and oceanic circulation alter physical properties in the ECS region, which can also affect the marine ecosystems of the ECS region (He et al., 2013; Xu et al., 2022; Wu et al., 2023). During a post-El Niño summer season, anomalous WNPAC transports warm and moist air from lower latitudes to East Asia, which increases regional precipitation (Li et al., 2021; Kwon et al., 2005) and thus river discharge, carrying abundant nutrients into the Yellow and East China Seas (YECS) region (Shi and Wang, 2012; Beardsley et al., 1985; He et al., 2013; Wu et al., 2023). This augmented river discharge impacts the coastal ecosystems of eastern China and the western part of the Korean peninsula, potentially triggering anomalous phytoplankton blooming (He et al., 2013; Yamaguchi et al., 2012; Park et al., 2015). In addition, during this period, anomalous southwesterly winds drive more nutrient-rich water from the South China Sea (SCS) into the ECS through the TS (Zhang et al., 2015; Huang et al., 2015). The recent increasing trend in surface chlorophyll-a in the ECS is correlated with nutrient concentrations without an increase in river discharge (He et al., 2013). In addition, increased river discharge can cause anomalous phytoplankton blooming after strong El Niño cases (Wu et al., 2023). These results suggest that various processes can induce enhanced phytoplankton blooming during post-El Niño summers.

The WNPAC generated during the El Niño period not only increases riverine flows and nutrient transport from TS but also accompanies anomalous cyclonic circulation over the EAMS region by the atmospheric Rossby wave propagation. This induces Ekman upwelling (EKU) in the ECS region, transporting nutrients from

the oceanic sub-surface to the surface layer, and fostering anomalous phytoplankton blooming. Together with
nutrient inputs via river discharge and the TS, this upwelling-driven mechanism could further amplify productivity
during these periods. An improved understanding of the various complex mechanisms in conjunction, it will allow
us to more accurately estimate the magnitude of phytoplankton blooms in the ECS region during post-El Niño
summers. This could have implications for marine resource management and fisheries in many neighboring
countries.

64            Here, we aim to comprehensively understand the variability of phytoplankton concentrations in the ECS
region during the post-El Niño summers. We investigate all of the phytoplankton blooming mechanisms: inland
riverine inflow and buoyancy upwelling by river discharge, nutrient inputs from the TS, and wind-driven oceanic
sub-surface upwelling and quantify the relative contributions of these three mechanisms. The climate model long-
term simulation and observational data support these findings, highlighting the concurrent roles of both
mechanisms in enhancing phytoplankton blooming.

## 2. Data and Methods

### 2.1. Reanalysis & Observation data

We used the ocean colour satellite data from the European Space Agency Climate Change Initiative Version (ESA-CCI) for chlorophyll-a data, serving as a proxy for phytoplankton biomass, covering 25-year periods from 1998 to 2022 (Sathyendranath et al., 2019). El Niño events were identified using Extended Reconstructed Sea Surface Temperature version 5 (ERSSTv5; Huang et al., 2017) Sea Surface Temperature (SST) data from National Oceanic and Atmospheric Administration (NOAA), with anomalies greater than 1 standard deviation during the winter season (December – January – February; DJF) index in the Nino3.4 (5°S-5°N / 170° W-120°W) region. Furthermore, to examine atmospheric circulation such as wind, precipitation, and geopotential height (GPH) changes, we analyzed the re-analysis data from the National Centers for Environmental Prediction reanalysis version 2 / the National Center for Atmospheric Research (NCEP2 / NCAR; Kanamitsu et al., 2002). Lastly, wind stress ($\tau$) and wind stress curl ($\mathrm{Curl}_{(\tau)}$; Kessler, 2006) are calculated followed as Eqs. (1)-(2) with typical value for drag coefficient ($C_D = 0.0015$; Trenberth et al., 1990; Wyrtki and Meyers, 1976) and sea level air density ($\rho_{air} = 1.225 \, \mathrm{kg \, m^{-3}}$; Cavcar, 2000) with the square of zonal & meridional wind speed ($V^2$) from NCEP2 re-analysis data. In addition, we used Simple Ocean Data Assimilation (SODA), version 3 from 1980 to 2020 to investigate the ocean currents (Carton et al., 2018). We used nutrients (Phosphate; $PO_4$) data from Copernicus Marine Environment Monitoring Service (CMEMS) Biogeochemical hindcast (10.48670/moi-00019) at monthly timesteps from 1998 to 2023. We calculated meridional $PO_4$ transport by multiplying the $PO_4$ concentration and meridional ocean current velocity, which was vertically integrated up to 50 m as Eq. (3) for the period from 1998 to 2020, when both data are available.

$$\tau = \rho_{air} C_D V^2 \tag{1}$$

$$\mathrm{Curl}_{(\tau)} = \frac{\partial \tau_y}{\partial x} - \frac{\partial \tau_x}{\partial y} \tag{2}$$

$$\int_{10m}^{50m} PO_4 \cdot v \; current \; velocity \; dz \tag{3}$$

### 2.2. Model data

We used the present climate-based (1990-year atmospheric $CO_2$ concentration level; 353 parts per million (ppm)) long-term (1,000 years) simulations of the Geophysical Fluid Dynamic Laboratory (GFDL) - CM2.1 Earth System Model (ESM) fully coupled with the marine ecosystem model TOPAZv2 (Tracers of Ocean Phytoplankton with Allometric Zooplankton Version 2; Dunne et al., 2013). The model incorporates various external nutrient inputs such as atmospheric deposition, river nitrogen (N) input, and river inputs of dissolved carbon, alkalinity, and lithogenic material; however not includes river phosphorus (P) inputs (Dunne et al., 2013). The growth rate of phytoplankton is computed using a function of chlorophyll to carbon ratio and limited by

various factors in TOPAZ (Dunne et al., 2010). Nutrient limitation terms are determined by minimum limitation
values among macronutrients & micronutrients (Fe, $PO_4$, $Si(OH)_4$, $NO_3 + NH_4$).

104       Surface values in the model data were averaged to 30-meter depths, with spatial grid and temporal scale

set at 1° x 1° grid and monthly mean data, respectively. All anomalies are calculated by removing the seasonal
cycles and the linear trend. El Niño events were defined in the same way as in the observation data. Statistical
significance was calculated using the non-parametric bootstrap method, with 10,000 random samples replacing
as many numbers of El Niño cases in observation and model results, respectively.

## 3. Results

### 3.1. Runoff & TS-transport-driven blooming mechanisms - Insight from Observations

Figure 1a displays the composite maps illustrating the spatial distribution of surface chlorophyll-a (SCHL) anomalies during the post-El Niño summer season (June-July-August; JJA). The pronounced anomalous SCHL blooming is observed from the Yangtze River Estuary (YRE) to the southern part of the Korean Peninsula. It has consistent patterns with the dispersal of strong river discharge from the YR (Yamaguchi et al., 2012; Park et al., 2015). In addition to the estuary of the YR region, broad, weak diagonal patterns of positive signals are also observed extending from northeastern Taiwan to the Korean Strait. This is in agreement with the relationship between Yangtze River Discharge (YRD) and SCHL in Yamaguchi et al. (2012) which has a lagged correlation from June to August and extends over a wide region from Jeju Island to the Korean Strait.

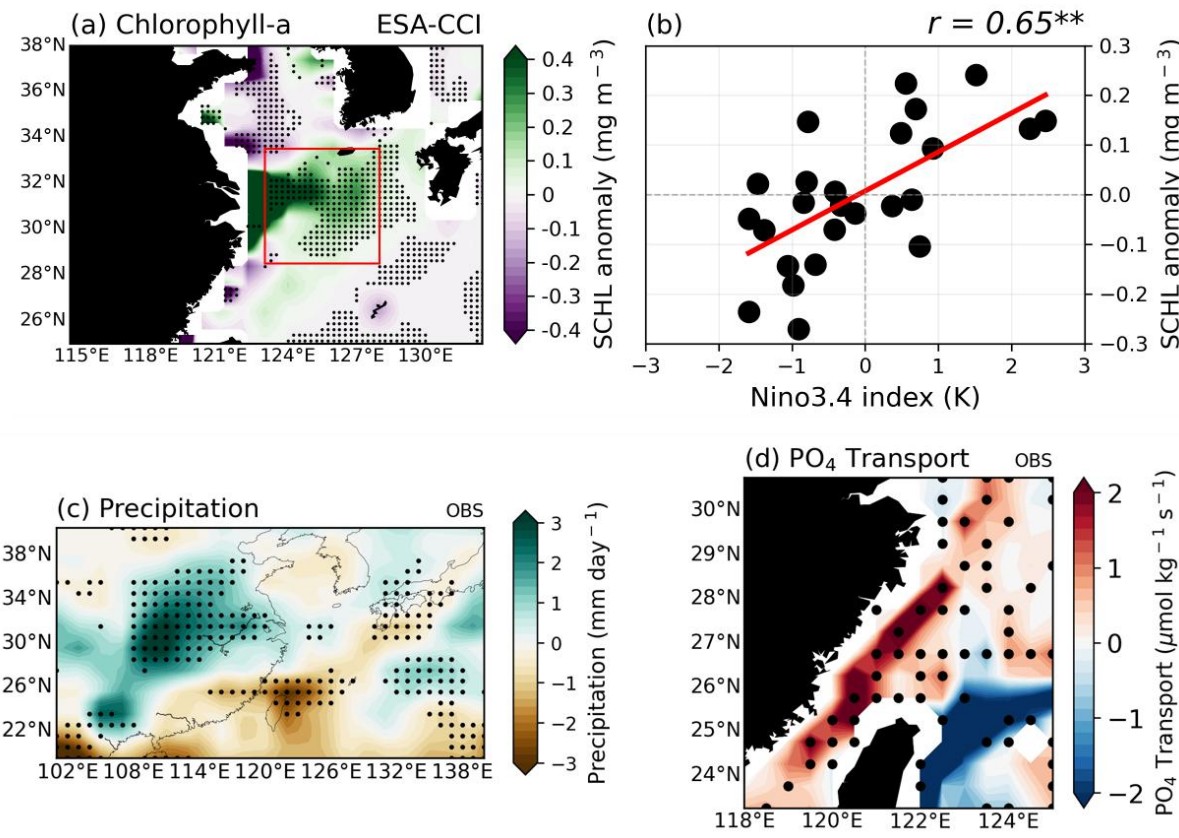

**Figure 1.** Composite maps exhibiting the 3-month average of **(a)** surface Chlorophyll-a (SCHL) anomalies using satellite data during the post-El Niño summer season (June-July-August; JJA). **(b)** The relationship between the Niño3.4-DJF index which indicates the strength of El Niño mature phase and area-averaged SCHL anomalies over the research target area during the post-El Niño summer season (Red box in Fig. 1a; 28.5°N-33.5°N / 123°E-128°E). **(c)** Precipitation anomalies for post-El Niño summer season of all El Niño cases, using re-analysis and satellite data. **(d)** Meridional $PO_4$ transport anomalies through the Taiwan Strait (TS) for post-El Niño summer season of all El Niño cases in reanalysis. All the black dots indicate where the responses are significant at the 90% confidence level by using the bootstrap method. ** marks indicate statistically significant correlation at the 99% confidence level.

During the post-El Niño summers (JJA(1)), the lagged correlation between the Nino3.4 D(0)JF(1) index and the SCHL anomalies in the ECS region (28.5°N-33.5°N, 123°E-128°E; target region indicated as red box in Fig. 1a) indicate significant positive relationship (r = 0.65**) at the 99% confidence level (Figure 1b). The anomalous phytoplankton blooms during the summers following El Niño events can be attributed to increased riverine discharge from the YR, driven by enhanced precipitation, and the substantial nutrient supply through the TS from the South China Sea (SCS). In the El Niño years, the intensification of southwesterly winds by the Western North Pacific (WNP) anticyclone flow brings in warm and humid air from lower latitudes. This, in turn, leads to anomalous rainfall in the East Asian region (Figure 1c). Given the strong association between the variability of the YRD and precipitation, it is reasonable to anticipate that an increase in riverine input corresponds to increased precipitation (Park et al., 2015). In addition to the direct riverine nutrient input, enhanced river discharge can facilitate nutrients to the ECS region through buoyancy upwelling (Chen, 2008; Chen et al., 2003; Chen, 2000; Hill, 1998). As the freshwater plume outflow surpasses the incoming discharge, a pronounced horizontal density gradient develops along the plume boundary. This gradient creates a pressure difference, which in turn drives a compensatory circulation to maintain the water balance. Essentially, the pressure gradient forces the denser, nutrient-rich water from below to rise along the plume boundary, thereby upwelling into the surface layer. Consequently, the nutrient-rich Kuroshio subsurface waters ascend along the ECS shelf edge, providing essential nutrients. This buoyancy-driven upwelling occurs independently of wind conditions, driven primarily by the physical properties of the subsurface waters in response to the enhanced river discharge.

Additionally, the WNP anticyclonic circulation and its curl intensify the sea level gradient of the northwestern-southeastern regions over the SCS and thus amplify water transport through the TS, delivering nutrient-rich SCS water into the ECS region during post-El Niño summers (Figure 1d; Huang et al., 2015; Zhang et al., 2015; Chen et al., 2015). The influx of nutrient-rich waters, particularly $PO_4$, further facilitates marine phytoplankton in the ECS region.

To comprehensively understand the mechanisms driving phytoplankton blooms in the ECS during the summer season following the ENSO mature phase, we first re-assessed two major processes suggested by several previous studies—the runoff-driven blooming mechanism and the TS-transport-driven blooming mechanism—as few studies have directly compared them in the context of ECS phytoplankton blooms. In the runoff-driven mechanism, robust positive precipitation anomalies centered in the southeastern part of China (where the YR flows) lead to intensified rainfall and consequently increase in river discharge. This enhanced river discharge delivers additional nutrients to the ECS, not only through direct river inflow but also via the entrainment of subsurface waters by buoyancy upwelling induced by the out-flowing river plume (Runoff-driven mechanism). The TS-transport-driven mechanism is marked by positive meridional $PO_4$ transport signals, indicating substantial nutrient flux into the ECS through the TS from the SCS during post-El Niño summers. This enhanced nutrient supply can significantly contribute to anomalous SCHL blooms in the ECS, highlighting the critical role of TS inflow in influencing phytoplankton growth (TS-Transport-driven mechanism).

**3.2. Runoff & TS-transport-driven blooming mechanisms - Insight from Model**

We further analyzed a long-term simulation using the GFDL-CM2.1 ESM with a fully coupled biogeochemical model to understand various phytoplankton blooming mechanisms. In our target area, the ECS region, climatologically, there is a strong influx of riverine runoff from the YR, which carries elevated Nitrate (N) concentrations (Kim et al., 2011; Moon et al., 2021; Wang et al., 2003). Under these marine environmental conditions, P is relatively constrained, and phytoplankton growth is primarily controlled by changes in P concentrations. Although the GFDL-CM2.1 ESM does not simulate P inputs from riverine inflows, it does account for P supply through the dynamic ascent of subsurface waters driven by buoyancy upwelling resulting from intensified river discharge. Therefore, beyond analyses based on observational and reanalysis data, long-term model simulations enable us to comprehensively understand the phytoplankton blooming mechanisms in the ECS during the following summers of El Niño events.

Prior to delving into further analyses, we verified whether GFDL-CM2.1 ESM adequately simulates the nutrient limitation in the target region ($26.5°N$-$32.5°N$, $122.5°E$-$128.5°E$; target region in the model), with strong P limitation observed in the YRE region and stronger N limitation appearing getting farther out (Figure 2). We confirmed the nutrient limitation by separating the target region into the YRE and the extended YRE region (red & blue box in Figs. 2e-f). In Figs. 2a-b, there is a relatively weak positive correlation (r = 0.25**) between surface $NO_3$ and SCHL anomalies in the YRE region, while a higher significant positive correlation (r = 0.72**) in the extended YRE region, indicating that N limitation begins to prevail. On the other hand, Figures 2c-d show a significantly strong positive correlation (r = 0.88**) between surface $PO_4$ and SCHL anomalies in the YRE region and no correlation (r = -0.03) over the extended YRE region. In particular, the partial correlation between the surface $PO_4$ and SCHL anomalies after removing the effect of $NO_3$ in the YRE region is almost unchanged at 0.86**, while the opposite effect is even lower at 0.1**. The spatial distribution of partial correlations shows that the effect of surface $NO_3$ on SCHL anomalies after removing the effect of surface $PO_4$ is stronger in the extended YRE region. On the other hand, the effect of surface $PO_4$ on SCHL anomalies after removing the influence of surface $NO_3$ is very strong not only in the YRE region but also over the YECS and Bohai Sea (Figures 2e-f). These support the dominant P limitation in the YRE region, implying that the strong P limitation in the YRE region is applied as well as the observations. These findings indicate the GFDL-CM2.1 ESM's ability to simulate the relationship between phytoplankton growth and nutrients in the ECS region.

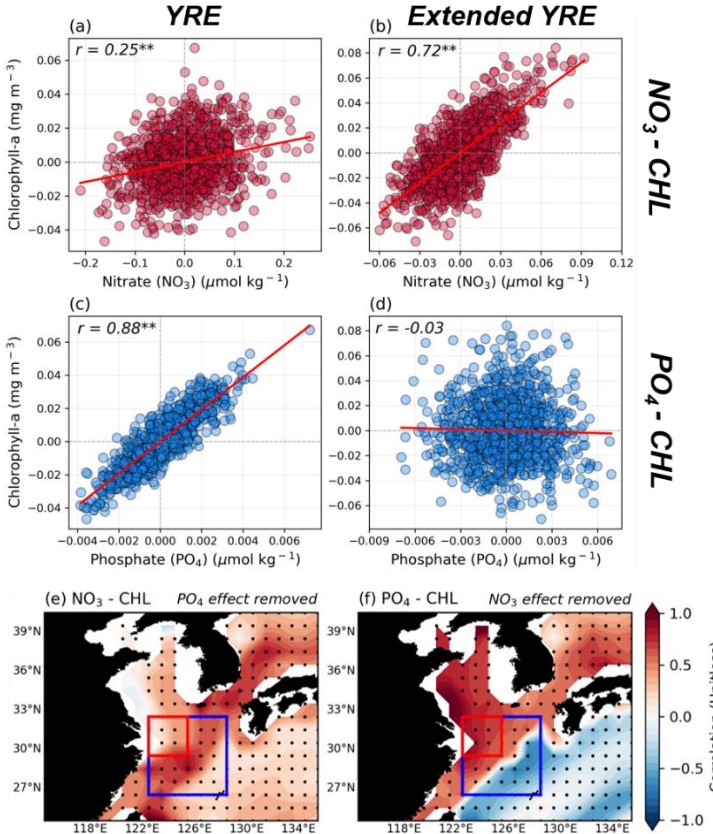

194

**Figure 2**. **(a-d)** The relationship between area-averaged surface nutrients and SCHL anomalies across the target area. The upper panels (a-b) focus on the relationship between surface nitrate ($NO_3$) and SCHL anomalies in the Yangtze River Estuary (YRE) [Left Panel; Fig. 2a] and extended YRE [Right Panel; Fig. 2b] in model results. The lower panels (c-d) indicate the relationship between surface phosphate ($PO_4$) and SCHL anomalies in YRE [Left Panel; Fig. 2c] and extended YRE [Right Panel; Fig. 2d] within the ECS region respectively. **(e-f)** Partial correlation distribution of the effect of surface $NO_3$ on SCHL anomalies after removing the effect of surface $PO_4$ and opposite case respectively. The red box and blue box with the excepted red box represent the YRE and Extended YRE regions respectively.

Figures 3a-c present the composites of SCHL anomalies, surface $NO_3$, and surface $PO_4$ anomalies during the post-summer season following the mature phase of El Niño from all El-Niño cases in the GFDL-CM2.1 ESM model over a total period of 1,000 years. Unlike the satellite-observed SCHL anomalies shown in Fig. 1a which are concentrated in the YRE region and gradually weakened outwards, the model results depict a diagonal band pattern extending from the southeastern part of China to the Korean Strait (Figure 3a). This anomalous SCHL pattern is somewhat consistent with the weak rightward diagonal pattern shown in Fig. 1a. Surface $NO_3$ anomalies in Fig. 3b exhibit strong signals primarily in the YRE region, akin to the SCHL anomalies observed in Fig. 1a. This similarity can be attributed to the ESM's incorporation of N inputs from riverine runoff, effectively capturing the observed patterns. In contrast, surface $PO_4$ anomalies (Figure 3c) show a diagonal shape with a positive signal from the northeastern part of Taiwan to the Korean strait. Given that the ESM does not include P input from riverine sources, these anomalous $PO_4$ patterns may result from nutrient supply through the TS. Additionally, nutrient entrainment from subsurface waters via buoyancy upwelling induced by enhanced river discharge could further support.

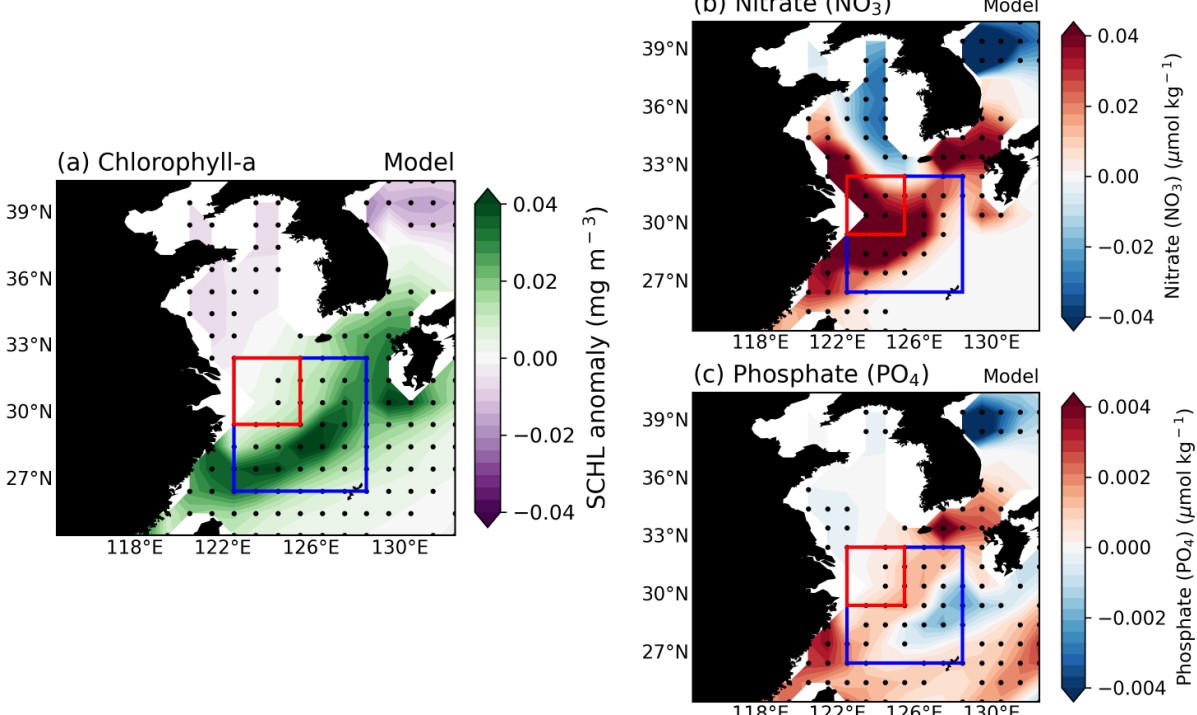

Figure 3. (a-c) Composite maps illustrate the SCHL, surface $NO_3$, and surface $PO_4$ distribution of post-El Niño summer season over the East China Sea (ECS) for all El Niño cases (176 years) which were selected in GFDL-CM2.1 ESM results. All the black dots indicate where the responses are statistically significant at the 95% confidence level, determined using the bootstrap method.

We examined the relationship between the DJF Niño3.4 index and SCHL anomalies in the ECS region, showing a significant positive correlation (r = 0.51**) at the 99% confidence level (Figure 4a). From the long-term integrated ESM results, we identified a total of 176 El Niño cases. Most of these cases result in positive anomalous SCHL blooming, however, about 25% of the total El Niño cases exhibit negative SCHL anomalies. To identify the processes responsible for the differing blooming outcomes, we divided the El Niño cases into two groups based on the magnitude of anomalous SCHL blooming in the target region. The Strong Blooming (SB) group, comprising the top 30 cases exhibiting strong blooming colored in reds, and the Non-Blooming (NB) group, comprising the bottom 30 cases colored in blues (Figure 4b).

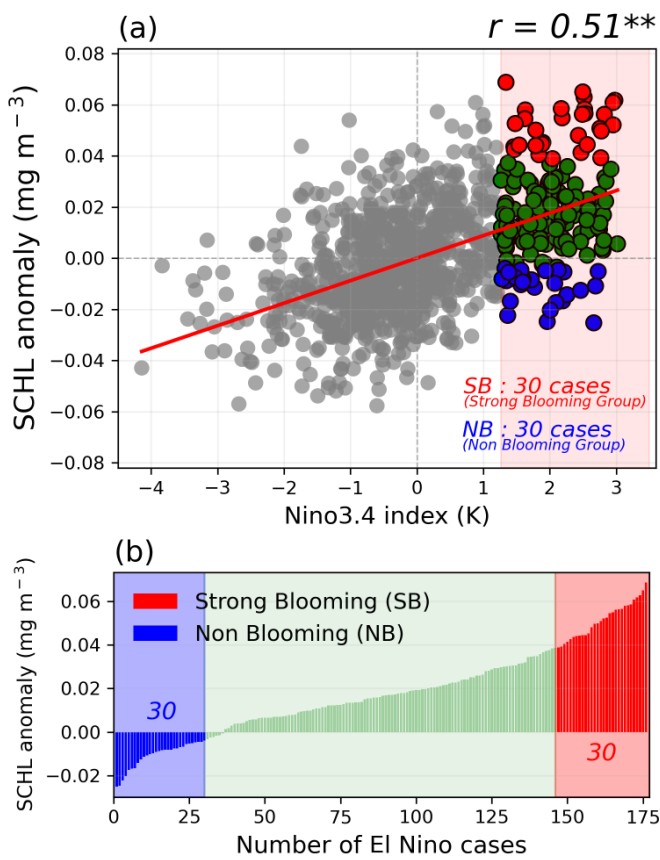

**Figure 4. (a)** The relationship between Niño3.4 DJF index and area-averaged SCHL anomalies over the target region in GFDL-CM2.1 Earth System Model (ESM) during post-El Niño summers**.** Gray dots indicate the Non-El Niño cases and colored dots indicate the El Niño cases. **(b)** Categorization of all El Niño cases by the magnitude of SCHL blooming in the target region of the ECS during post-El Niño summers. The top 30 and bottom 30 cases, distinguished by SCHL anomaly magnitudes over the target region, colored in red and blue are grouped into the Strong Blooming El Niño (SB) group and the Non-Blooming El Niño (NB) group, respectively.

The composite map of SCHL anomalies during post-El Niño summers of the SB group reveals pronounced anomalous blooming across the ECS region, while the NB group is characterized by negative SCHL anomalies (Figure 4b, Figs. S1a-c). Surface nutrient distributions from the composite maps also show contrasting results between the two groups during post-El Niño summers (Figs. S1d-i). To investigate the runoff-driven phytoplankton blooming mechanism, as discussed in previous studies and above, we analyzed the composite patterns of precipitation and runoff anomalies during post-El Niño summer season by comparing the two groups (Figure 5). In terms of precipitation patterns, the SB group exhibits a broad positive rainfall band extending over the entire East Asia region, from central China to Japan. In contrast, the NB group shows only weak positive precipitation anomalies, confined to a narrow region of China, Taiwan, and the southern part of Japan. Similarly, the runoff anomaly patterns display positive signals centered on the YRE region and the northern part of the YRE region for both groups, with significance observed only in the SB group. The NB group, however, exhibits only weakly significant positive patterns around the southern part of the YRE region. Moreover, the relationship

between runoff anomalies and the SCHL anomalies in the target region demonstrates a significantly positive correlation (r = 0.59**) at the 99% confidence level, indicating that the runoff-driven mechanism is well simulated in the model (Figure 5g).

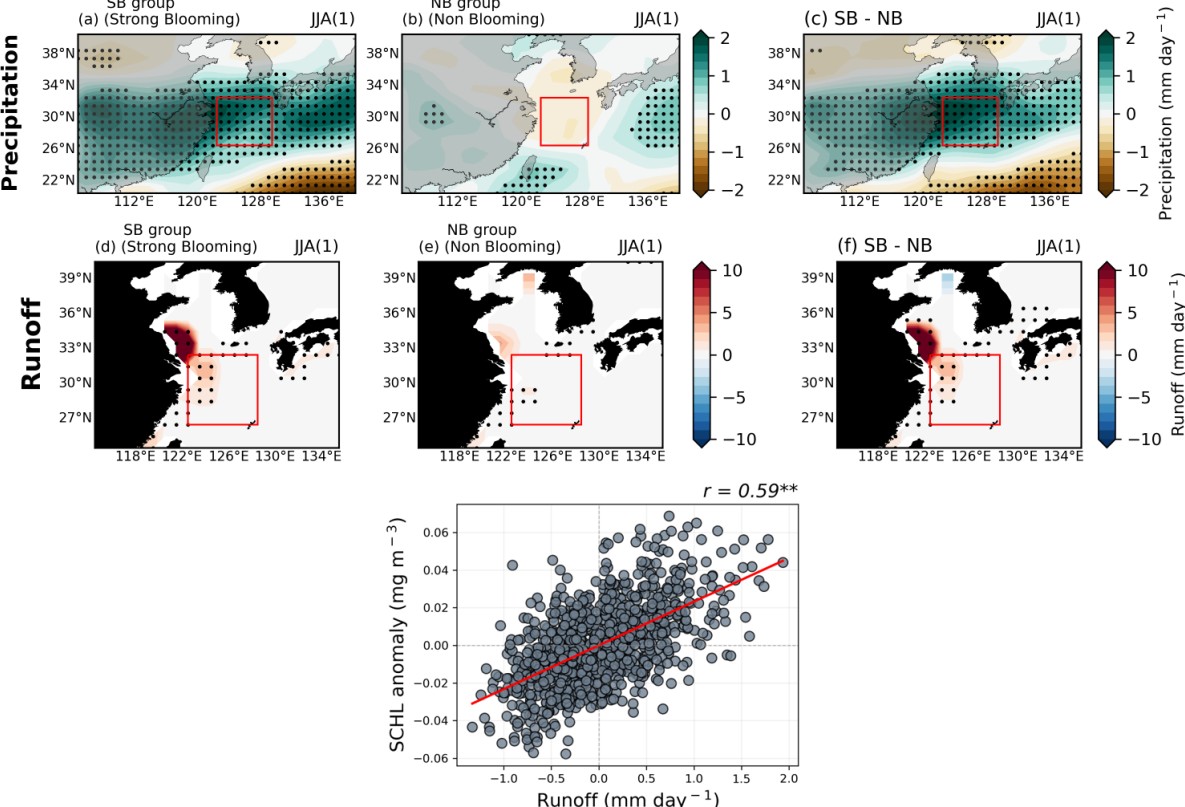

**Figure 5. (a-c)** Composite maps exhibiting precipitation anomalies during post-El Niño summer season for both groups, as well as the difference between the two groups. **(d-f)** Similar to Figs. 5a-c, these maps demonstrate a composite map of runoff anomalies during post-El Niño summer season for both groups, as well as the difference between the two groups. **(g)** The relationship between area-averaged runoff and SCHL anomalies over the target region. All figures show the GFDL-CM2.1 model results.

We also investigated the TS-transport-driven blooming mechanism, which is well known for its significant role in nutrient supply through the TS. We analyzed the meridional $PO_4$ transport by comparing two groups as the same as Fig. 5, and found remarkably distinct differences between them (Figure 6a-c). Notably, within the TS, a strong positive (negative) signal was identified in the SB (NB) group. For the SB group, the nutrient flux exhibits a significant positive signal reaching the target region, potentially supplying large amount of nutrients and making a substantial contribution to SCHL blooming in the ECS region. Furthermore, a positive correlation of 0.68—higher than the runoff-driven mechanism—was identified between the anomalous SCHL blooming magnitude in the target region and the meridional $PO_4$ transport index indicated by the blue diagonal box in the TS region (Figure. 6d). This underscores the critical role of the TS-transport-driven mechanism in

269    fostering nutrient into the ECS region.

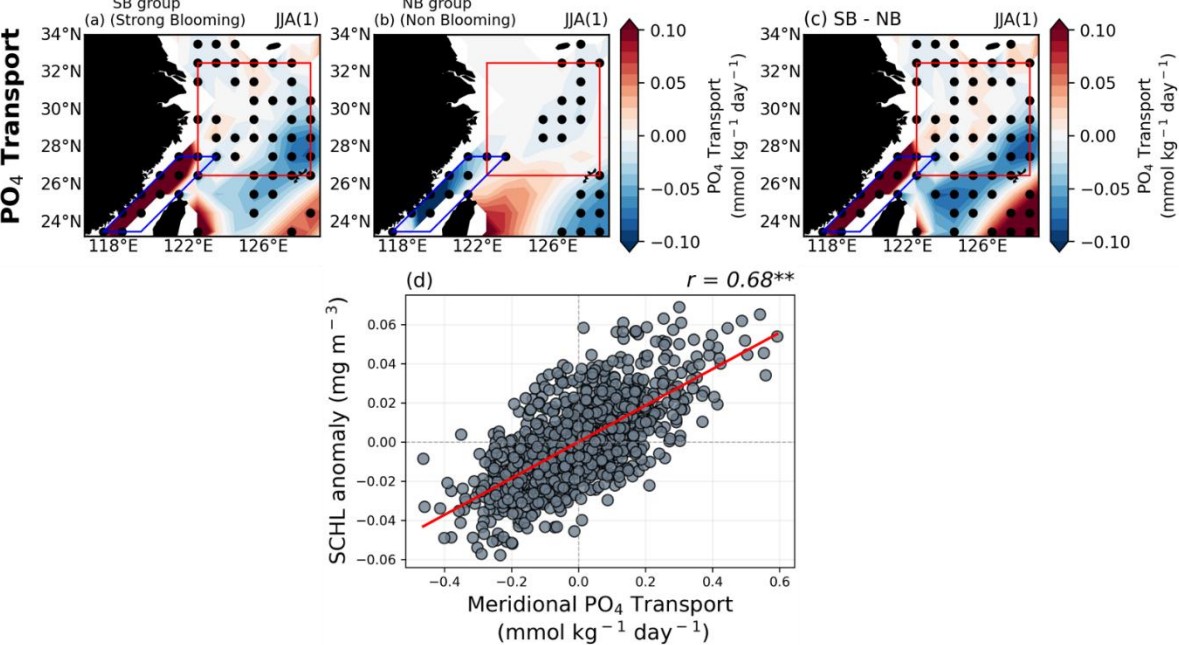

270

**Figure 6. (a-c)** Composite maps exhibiting meridional PO$_4$ transport anomalies during post-El Niño summer season for both groups, as well as the difference between the two groups. **(d)** The relationship between area-averaged meridional PO$_4$ transport and SCHL anomalies over the target region. All figures show the GFDL-CM2.1 model results.

### 3.3. Upwelling-driven blooming mechanism - Insight from Model

In addition to runoff-driven and TS-transport-driven mechanisms, there was a distinctive difference in the magnitude of upwelling between the two groups. Figure 7 shows the composite map of Ekman Upwelling (EKU) anomalies during post-El Niño summer season. For the SB group, a significantly positive EKU pattern dominates the ECS region, with significantly negative EKU distribution in the WNP region, far south of Japan (Figure 7a). On the other hand, the NB group does not exhibit any significant EKU anomaly patterns in the target region, with significant positive EKU patterns over the southern part of Japan and negative EKU anomalies in the WNP but without significance (Figure 7b). The difference between the two groups highlights distinct EKU anomaly patterns in the target region (Figure 7c). The intensified EKU in the target region facilitates the supply of abundant nutrients from the subsurface layer to the surface layer, thereby enhancing phytoplankton growth. This is supported by the significantly positive correlation (r = 0.47**) at the 99% confidence level between EKU anomalies and SCHL anomalies in the target region, indicating that EKU can significantly contribute to anomalous phytoplankton blooming (Figure 7d).

EKU is primarily generated by cyclonic atmospheric circulations. The presence of robust upwelling in the ECS region signifies the formation of cyclonic circulation, i.e. the EKU and wind stress curl (WSCL) are fully positively correlated (Fig. S2a). During the El Niño mature phase, suppressed convection in the western Pacific

induces the Western North Pacific Anticyclone (WNPAC) in the lower-troposphere structure (Gill, 1980; Matsuno,
1966; Xie et al., 2016; Chowdary et al., 2019). Moreover, the WNPAC leads to cyclonic circulation in the
northwestward ECS region via low-level Rossby wave energy propagation (Wang et al., 2000). This sequence of
wave patterns in the lower troposphere can generate cyclonic circulation in the ECS region during post-El Niño
summer seasons.

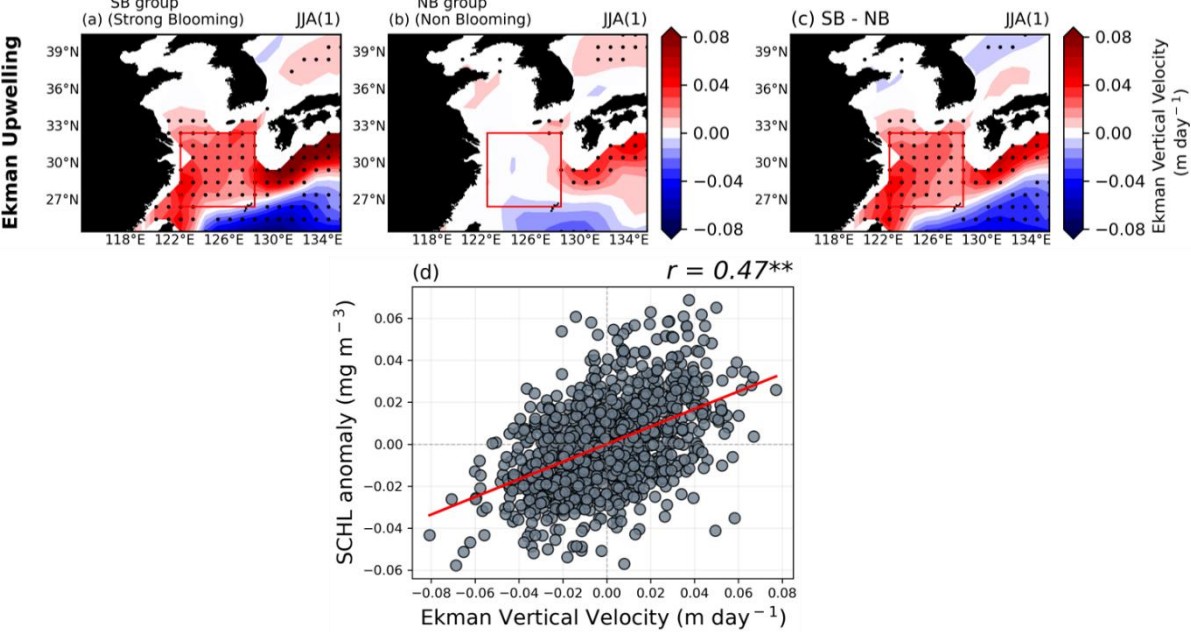


**Figure 7. (a-c)** Composite maps exhibiting Ekman Upwelling (EKU) anomalies during post-El Niño summer
season for both groups, as well as the difference between the two groups. **(d)** The relationship between area-
averaged EKU and SCHL anomalies over the target region. All figures show the GFDL-CM2.1 model results.

302        Given the Rossby wave teleconnection pattern, it is expected that the EKU response can be related to
the strength of the WNPAC. Figure 8 illustrates the evolution of geopotential height (GPH) differences with
850hPa wind (Left panels) and WSCL anomalies (Right panels) between two groups in the WNP across three
seasons (D(0)JF(1)-MAM(1)-JJA(1)) following El Niño mature phase. In the SB group, the WNPAC and North
Pacific Cyclone (NPC) are prominently stronger, positioned over the Philippines and North Pacific, respectively
(Figure 8a). These atmospheric circulations result in anomalous positive WSCL around the Korean Peninsula,
Japan, and East Sea, while negative WSCL is observed in southeastern China (Figure 8d). As the seasons progress,
the WNPAC migrates northeastward, becoming more pronounced in the SB group compared to the NB group,
continuing into the subsequent summer (Figures 8b-c). Positive WSCL anomalies begin to emerge in the ECS
region from El Niño decaying spring season, coinciding with the developed WNPAC (Figure 8e). By summer, the
WNPAC in the SB group is intensely and broadly developed, dominating the WNP, leading to a stronger cyclonic
circulation in the ECS region and enhanced wind-driven EKU due to positive WSCL (Figure 8c and Figure 8f).
The correlation between WSCL and the WNPAC index with slightly broader than observations (13.5°N–26.5°N /
124.5°E–160.5°E; which is slightly broader than observations) calculated from GPH anomalies within the red box

in Fig. 8c exhibits a significantly positive relationship (r = 0.45**) at the 99% confidence level. This implies that the development of a stronger WNPAC may lead to the generation of anomalous positive WSCL, prompting upwelling and facilitating anomalous phytoplankton blooming during post-El Niño summers.

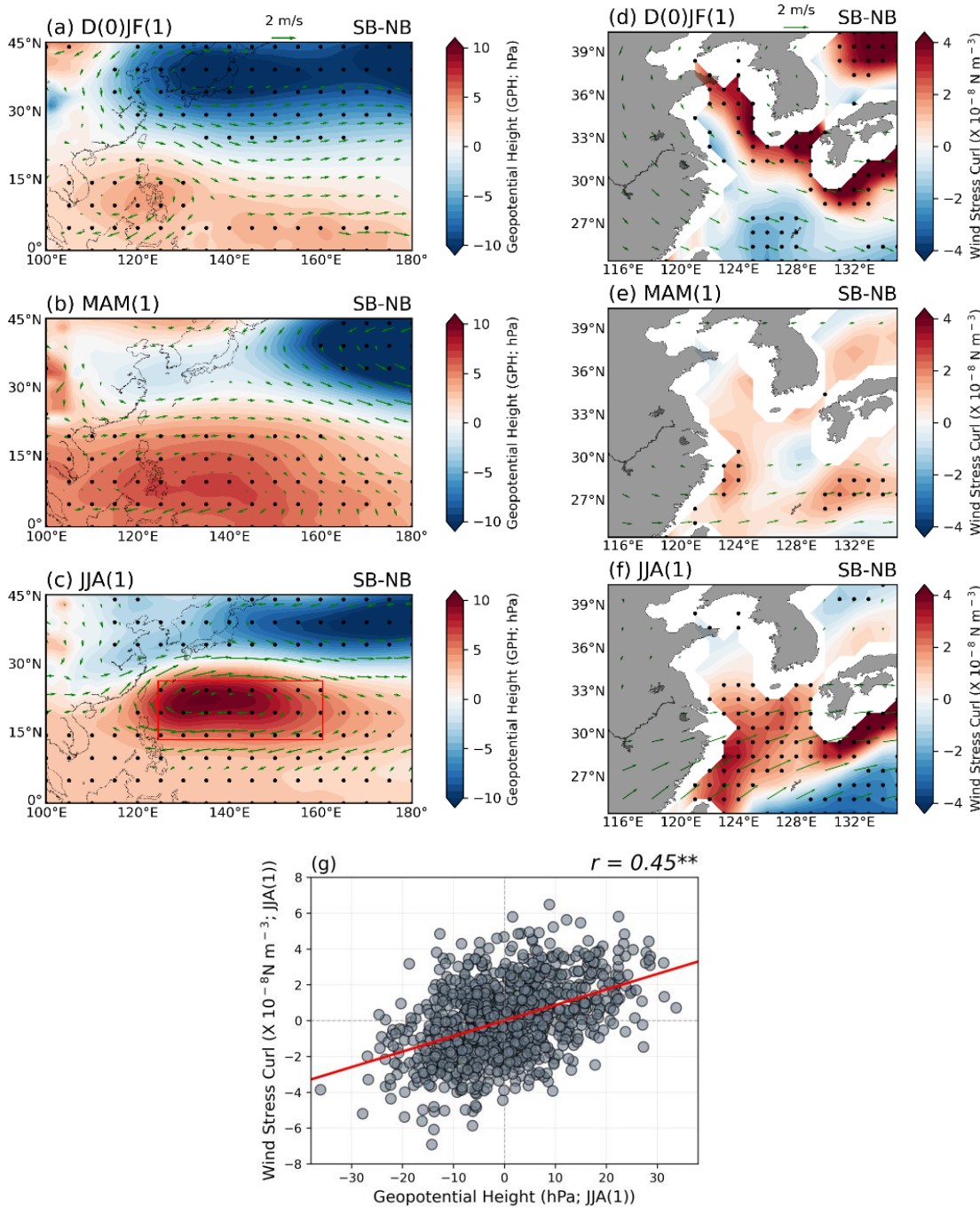

**Figure 8.** Evolution map of El Niño composite of (a-c) geopotential height (GPH; Left panels) and (d-f) Wind Stress Curl (WSCL; Right panels) with 850hPa wind anomalies (vectors) from El Niño mature phase to decaying summer season for the differences between two groups. All the black dots indicate where the responses are statistically significant at the 95% confidence level, determined using the bootstrap method. (g) The relationship

between area-averaged Western North Pacific (WNP) GPH anomalies (highlighted as a red box in Fig. 8c) and Wind Stress Curl (WSCL) anomalies over the target region during the summer season. All figures show the GFDL-CM2.1 model results.

So far, we have conducted a comprehensive evaluation and analysis of the complex mechanisms driving phytoplankton blooms in the ECS during the summers following the El Niño events. In addition to the runoff- and TS-transport-driven mechanisms suggested in previous studies, we introduced the Ekman upwelling-driven blooming mechanism. Subsequently, we quantified the relative contributions of these three mechanisms based on regression methods using (Eq. 4) as follows:

$$\frac{dChl}{dNino3.4}(\alpha) = \frac{\partial Chl}{\partial Runoff} \times \frac{dRunoff}{dNino3.4} + \frac{\partial Chl}{\partial TS-transport} \times \frac{dTS-transport}{dNino3.4} +$$

$$\frac{\partial Chl}{\partial Ekman-Upwelling} \times \frac{dEkman-Upwelling}{dNino3.4} + residual \qquad (4)$$

Before quantitatively assessing the relative contributions of each mechanism, we evaluated potential multicollinearity among the three mechanisms by calculating the Variance Inflation Factor (VIF). The VIF values for the three mechanisms—Runoff (1.265), TS-transport (1.08), and Ekman Upwelling (1.214)—ranged from 1.08 to 1.265, indicating minimal multicollinearity. VIF values below 3 are typically considered negligible multicollinearity, suggesting that three mechanisms are statistically independent (Kock and Lynn, 2012; Kim, 2019). Firstly, we normalized all variables and conducted the multiple regression with respect to SCHL index in the target region as the dependent variable and three mechanism indices as independent variables. Secondly, we calculated each term by multiplying the linear regression coefficient between Nino3.4 index and each mechanism index by the corresponding multiple regression coefficient for each mechanism index. The combined effects of the three mechanisms and the residual term collectively account for the variation of the SCHL index in the ECS region with respect to the El Niño events. Therefore, each term indicates how changes in three mechanism induced by ENSO cycle (as indicated by the Nino3.4 index) affect on phytoplankton blooms in the ECS region. The results show the effect of runoff-driven mechanism and the TS-transport mechanisms to SCHL blooming intensity in the target region are comparable (Table 1). In the case of the Ekman upwelling mechanism, it accounts for about 40% to 47% of contributions comapred to the other two mechanisms, yet still exerts a meaningful influence as well.

**Table 1.** Relative contributions of three mechanisms to SCHL blooming in the target region.

| | $\alpha$ | Runoff | TS-transport | Ekman Upwelling |
|---|---|---|---|---|
| Contribution | 0.511 | 0.152 | 0.138 | 0.065 |

Additionally, we identified their spatial contributions to further elucidate the effects of each mechanism
within the target region (Figure 9). We conducted multiple regression for three mechanism indices at each grid
point of SCHL anomalies across the target region. Following the same way examined above, we calculated the
spatial contributions by multiplying the linear regression coefficents of each mechanism with repsect to the
Nino3.4 index. The results showed that the runoff-driven mechanism exhibited the strongest contribution at the
center of the target region, driven by direct riverine nutrient inflow (though the model doesn't simulate P input)
and nutrient supply through buoyancy upwelling induced by river discharge (Figure 9a). In contrast, the TS-
transport-driven mechanism significantly contributes along a southwest-northeast diagonal direction, aligning
with nutrient transport via the TS from the SCS (Figure 9b). Additionally, the newly proposed Ekman upwelling-
driven mechanism, while relatively lower in overall contribution, shows significant and uniformly distributed
impacts across the target region (Figure 9c). These results suggest that all mechanisms can affect comprehensively
to induce phytoplankton blooms in the ECS during post-El Niño summers. Therefore, considering all mechanisms
is essential for accurately predicting the intensity of phytoplankton blooms.

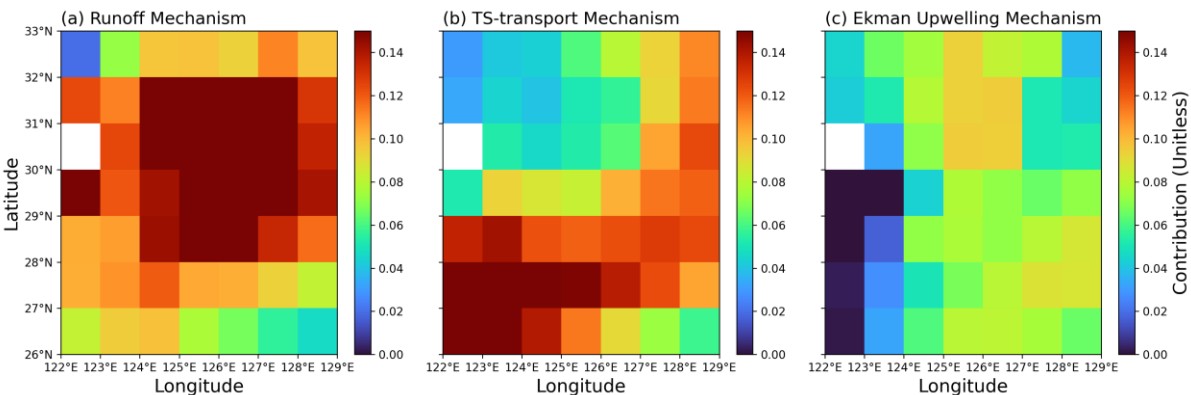


**Figure 9.** Relative contributions of three mechanisms to SCHL blooming in the target region during post-El Niño
summers **(a)** Runoff-driven mechanism **(b)** TS-transport-driven mechanism **(c)** Ekman upwelling-driven
mechanism. All figures show the GFDL-CM2.1 model results.

### 3.4. Upwelling-driven blooming mechanism - Insight from Observations

Lastly, we examined the upwelling-driven mechanism using reanalysis data for physical variables and
chlorophyll-a satellite data to verify its application to the real world. Figure 10a shows the composite map of GPH
anomalies during post-El Niño summers, using NCEP2 re-analysis data. To identify the typical WNPAC and
850hPa wind anomalies during post-El Niño summers, we defined a WNPAC index using the full period of
available NCEP2 re-analysis data (1979-2022; 44 years). The WNPAC index was defined as the area-averaged
(16.5°N-26.5°N, 130°E-155°E) GPH anomalies over the WNP region indicated by the red box in Fig. 10a. The
WNPAC pattern and anticyclonic 850hPa wind anomalies ranging from the South China Sea to the WNP, are well
represented in the re-analysis data. We calculated the WSCL using surface winds from the NCEP2 re-analysis

data, revealing significant positive WSCL (cyclonic circulation) anomalies over a large area of southeastern China (Figure 10b). Furthermore, we found a significant positive correlation (r = 0.56**) at the 99% confidence level between the WSCL index (red box in Fig. 10b) and SCHL anomalies in the target region (Figure 10c). Additionally, there is a significant positive correlation (r = 0.43*) at the 95% confidence level between the WNPAC index and the SCHL anomalies in the target region (Figure 10d). A strong positive relationship (r = 0.56**) at the 99% confidence level between the WNPAC index and WSCL in the target region was also observed (Figure 10e). Several instances of negative SCHL anomalies occurred despite positive anomalous rainfall (blue box in Fig. 10a; 28.5°N-33.5°N, 111°E-120°E). However, interestingly, cases exhibiting strong anomalous SCHL blooming mostly coincided with either strong WNPAC indices or robust cyclonic WSCL in the ECS region (Figure 10e). These remarkable associations among these variables suggest that the upwelling-driven blooming mechanism is indeed operating in the real world. Therefore, all of the mechanisms suggested in this study including upwelling-driven mechanisms must be considered together to fully understand the dynamics of phytoplankton blooming in the ECS region during post-El Niño summers.

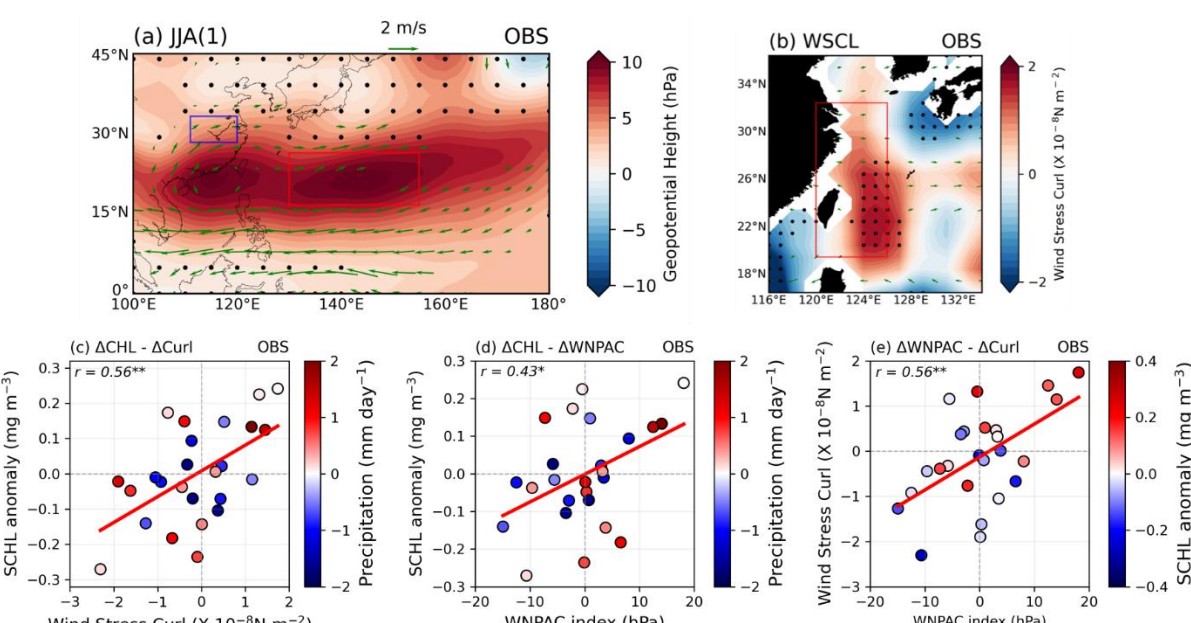

**Figure 10. (a)** The composite map exhibits the GPH anomalies over the WNP and 850hPa wind anomaly patterns during the post-El Niño summer season for the available period of re-analysis data (1979-2022; 44 years). Red and blue boxes indicate the area-averaged WNPAC index (16.5°N-26.5°N, 130°E-155°E) and precipitation index (28.5°N- 33.5°N, 111°E-120°E) respectively. Black dots indicate the insignificance at the 90% confidence level using the bootstrap method. **(b)** The composite map shows WSCL anomaly patterns and 850hPa wind anomaly patterns during post-El Niño summer season within the same period with Fig. 10a. The red box indicates the area-averaged WSCL index. Black dots indicate the significance at the 90% confidence level using the bootstrap method. **(c-d)** The relationship between SCHL anomalies in the target region and the WSCL index and the WNPAC index respectively. Colored scatters show the precipitation index over the eastern part of China. **(e)** The relationship between the WNPAC index and the WSCL index. Colored scatters show the magnitude of SCHL anomalies in the target region.

## 4. Discussions

This study investigates the comprehensive mechanisms of anomalous phytoplankton blooms in the ECS region during the summer following El Niño events. We re-evaluated the existing runoff-driven and TS-transport-driven phytoplankton blooming mechanism using the observational data and long-term integrated ESM results based on present climate conditions. Notably, the ESM we used does not simulate P riverine input, a crucial factor in the YRE region, our research target region. Despite the absence of P riverine inputs, significant positive phytoplankton blooms still emerged, we hence proposed an additional mechanism to trigger anomalous phytoplankton blooming during the summers following the El Niño mature phase.

Firstly, we sorted all El Niño cases selected from long-term climate simulation results based on the magnitude of SCHL blooming in the ECS region and classified the El Niño cases into two groups (SB & NB group). There were clear differences between the two groups in climatic factors such as precipitation, runoff, and meridional $PO_4$ transport along TS pathway which drive blooming mechanisms commonly. In the SB group, stronger and broader rainfall band and riverine inputs into the ECS region were significantly predominant. The rainfall band generated over East Asia during the summer season of post-El Niño is related to the development of the WNPAC, suggesting that the WNPAC is more potently advanced in the SB group. Moreover, in the SB group by the enhanced development of WNPAC, the anomalous southwesterly winds associated with the SCS drive a pronounced augmentation of northward nutreint-rich water transport through the TS.

In addition, distinct differences between the two groups were observed, evident in both the magnitude of EKU anomalies induced by atmospheric circulation and the WSCL anomaly patterns. Consequently, We found a more robust and expansive WNPAC that persisted and extended its influence until the summer season, impacting a broader area of East Asia in the SB group. This well-developed WNPAC triggers stronger wind anomalies in the target region, leading to active EKU with positive WSCL. It facilitates upwelling in the water column as the cyclonic atmospheric circulation, driven by the strong clockwise circulation over WNP, prevailed across the entire ECS region.

Lastly, we validated the upwelling-driven phytoplankton blooming mechanism elucidated by the model results using reanalysis data and satellite chlorophyll-a data. There was a significantly positive relationship between WNPAC and SCHL anomalies, as well as WNPAC and WSCL anomalies in the target region. Thus, depending on the development of WNPAC during post-El Niño summers, anomalous phytoplankton blooms can be triggered by a conjunction of complex mechanisms: runoff-driven accompanied by strong precipitation in the ECS region, and TS-transport-driven by enhanced nutrient-rich water transport from SCS to ECS region through TS, as well as upwelling-driven mechanism induced by positive WSCL and EKU.

We investigated that the more vigorous development and expansion of the WNPAC can influence the marine ecosystems of the ECS region with sufficient El Niño cases using ESM results. The intensity and extent of WNPAC development between the SB and NB groups were distinct, which means that anomalous phytoplankton blooming can be predicted as early as two seasons before it occurs during the El Niño mature phase. We found a significantly positive lagged relationship between the WNPAC index and blooming magnitude in the ECS region in both observations and ESM results (Figure 11a-d; Observations, Figure 11e-h; Model). There were

significant 2 seasons (D(0)JF(1) – JJA(1)) lagged positive correlations (r = 0.61**) and ESM results (r = 0.5**).
In addition, the case of a short-term, 1 season lagged relationship (MAM(1) – JJA(1)) has higher lagged positive
correlations in both observations (r = 0.69**) and ESM results (0.57**) at the 99% confidence level. These results
demonstrate that the magnitude of WNPAC during the El Niño mature phase can be a good predictor of the
magnitude of phytoplankton blooms in the ECS region during the following summers of El Niño events.

448       In our study, we used a global climate model to investigate how large-scale climate variability influences
oceanic biogeochemical processes. The model has a relatively coarse resolution of 1°×1° across the global domain,
which limits its ability to resolve small-scale eddies and coastal upwelling, potentially leading to an
underestimation of SCHL variability. Despite this limitation, the model reasonably captures the observed spatial
and temporal patterns of SCHL variability, allowing us to effectively examine the physical mechanisms driving
these variations. However, we acknowledge that the quantitative contribution of each physical process could be
resolution-dependent. Therefore, future studies using higher-resolution models would be valuable for providing a
more precise quantification of these processes.

456       This study relies on the results of present-climate model simulations, and there may be changes in the
pattern of WNPAC development due to changes in the ENSO teleconnection resulting from changes in the El
Niño mean state under global warming scenarios (Yeh et al., 2009; Shin et al., 2022; Yang et al., 2022; Kim et al.,
2024; Wang et al., 2023). Here, we primarily focused on the biological aspect of the phytoplankton blooming
mechanism and did not extensively explore the dynamic mechanisms driving differences in WNPAC development
intensity between the two groups. In Fig. S3, we identified differences in the decaying speed of the El Niño
strength between the two groups and the warmer SST anomalies in the Central Pacific (CP) region during El Niño
mature phase in the SB group. Therefore, further studies will be necessary to investigate the blooming magnitude
variations associated with different El Niño types such as CP and Eastern Pacific (EP) El Niño (Kug et al., 2009,
2010; Yuan and Yang, 2012). Moreover, Xie et al (2009) mentioned that the anomalous large-scale anti-cyclone
in the WNP during the summer season of the El Niño decaying is associated with the Indian Ocean (IO), named
the IO Capacitor theory. These dynamic aspects will require further detailed exploration in subsequent studies.
Additionally, quantitative analyses between all of the suggested blooming mechanisms through model
experiments will be necessary to provide a detailed understanding of the contribution of all mechanisms to
phytoplankton blooms during post-El Niño summer season.

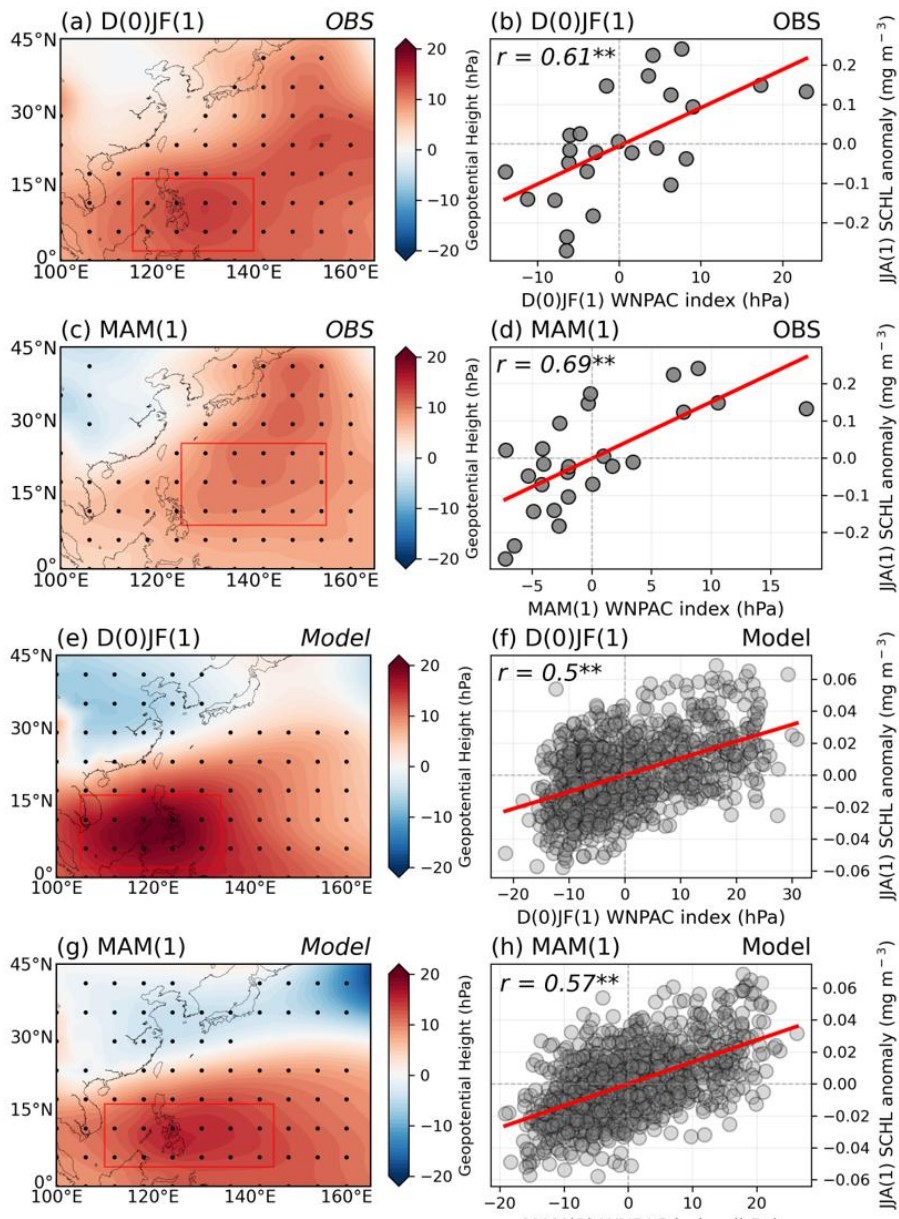


**Figure 11. (a-d)** Evolution maps indicate the distribution of GPH anomalies from the mature phase (D(0)JF(1)) of El Niño to the following spring season (MAM(1)) in the observations. Scatter plots show the lagged relationship between the area-averaged WNPAC index (red box in the left panels) during each period and the anomalous SCHL blooming magnitude in the target region during post-summers of the El Niño events in observations. **(e-h)** Same as (a-d), but with model results.

**Data availability**
The chlorophyll satellite observation data used in this study is available on https://esa-
oceancolour-cci.org. The ERSST (SST), wind, wind stress, GPH re-analysis data are provided
at https://psl.noaa.gov/ and nutrient data are provided at https://data.marine.copernicus.eu/.

**Code availability**
The computer codes that support the analysis within this paper are available from the
corresponding author upon request.

**Funding**
This work was sponsored by a research grant from the National Research Foundation of Korea
(NRF-2021M3I6A1086808), and supported by Research of Long-term Marine Environmental
Change in the Yellow Sea and East China Sea Using the Ieodo Ocean Research Station (Ieodo-
ORS) form the Korean Hydrographic and Oceanographic Agency (KHOA). This work was
also supported by Institute of Information & communications Technology Planning &
Evaluation (IITP) grant funded by the Korea government(MSIT) [NO.RS-2021-II211343,
Artificial Intelligence Graduate School Program (Seoul National University)]

**Ethical approval**
Not applicable

**Declaration of Competing interests**
The authors declare no competing interests.

**CrediT authorship contribution statement**
DGL compiled the data, conducted analyses, prepared the figures, and wrote the manuscript.
JSK designed the research and wrote the majority of the manuscript content. All the authors
discussed the study results and reviewed the manuscript.

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
