# Peer review of "Phytoplankton blooming mechanisms over the East China Sea"

_EGUsphere, 2024_

## Author Comment (AC3)

**Review of "Two different phytoplankton blooming mechanisms over the East China Sea during El-Niño decaying summers" by Lee et al.**

**Comment on egusphere-2024-3406 | Anonymous Referee #1:**

This manuscript discusses an important issue of international interest. However, it is premature because of major flaws.

The major flaw is the erroneous assumption that riverine input is the primary source of nutrients(line 27). Much of the Introduction and Methods concerns river input. However, the Kuroshio Intermediate Water is the primary source of nutrients for the East China Sea shelf. Many papers substantiated this notion. The authors used the results presented in Figs. 2 and 3 to prove the importance of riverine inputs of nutrients. Yet, Figs. 2 and 3 only provided a good correlation between chlorophyll and nutrients that mostly did not come from rivers. Instead, the upwelling of nutrient-rich Kuroshio subsurface waters provided most of the nutrients. Of course, upwelling could be induced by the buoyance effect caused by the river water(e.g., Chen 2008, Acta Oceanologica Sinica, 27,133, 2008). To summarize, the so-called runoff-driven blooming mechanism is not caused by riverine nutrients. Instead, it is caused by buoyancy-driven upwelling and vertical mixing. BTW, validation is needed to substantiate the modeled results presented in these two figures.

Another factor worth mentioning is that the outflow of the South China Sea is the primary source of Kuroshio waters entering the East China Sea. During El Nino years, there is a more substantial outflow of the more nutrient-rich SCS water(e.g., Chen et al., Deep-Sea Res. I, 103,13, 2015), which may enhance productivity on the ECS shelf.

As a final note, many of the figures are not labeled correctly. Figures 1, 2, 3, 4, 5, 6, 9, and 10 are all related to anomalies, not the actual values. The "p" values should also be provided.

**Response:** We thank the reviewer for constructive and valuable comments and suggestions on our results. We have made substantial revisions to the manuscript in response to the referee's key comments.

We sincerely appreciate referee #1 for the comment that nutrient supply from the entrainment of subsurface waters via dynamic buoyancy upwelling induced by great out-flowing river plume is an important influence on phytoplankton blooms in the ECS region as well. Obviously, we missed this physical process in the original manuscript. By following the referee's suggestions, we have added a clarification to the manuscript to state that the runoff-driven mechanism also includes nutrient supply from the subsurface water entrainment due to buoyancy upwelling caused by enhanced river discharge during the El Niño decaying summers in the ECS continental shelves as follows:

(L139-145): In addition to direct riverine nutrient input, enhanced river discharge can deliver nutrients to the East China Sea (ECS) region through buoyancy-driven upwelling (Chen, 2008; Chen et al., 2003; Chen, 2000; Hill, 1998). When the outflowing river plume becomes more substantial than the incoming river discharge, subsurface waters are upwelled to maintain water balance. This process allows nutrient-rich Kuroshio subsurface waters to ascend along the ECS shelf edge, supplying essential nutrients to the region. Unlike wind-driven upwelling, this buoyancy-driven upwelling is governed by the physical properties of the subsurface waters in response to intensified river outflow.

(L170-174): Although the GFDL-CM2.1 ESM does not simulate P inputs from riverine inflows, it does account for P supply through the dynamic ascent of subsurface waters driven by buoyancy upwelling resulting from river discharge. Therefore, beyond analyses based on observational and reanalysis data, long-term model simulations enable us to comprehensively understand the phytoplankton blooming mechanisms in the ECS during the following summers of El Niño events.

Additionally, we have substantially revised the manuscript in response to the referee's comments. As the reviewer suggested, we discussed the role of nutrient transport through the Taiwan Strait (TS) (Huang et al., 2015; Zhang et al., 2015; Chen et al., 2015) by adding new analyses. In the revised manuscript, we have extensively addressed this by incorporating the nutrient transport mechanism through the TS (TS-transport-driven mechanism). Using nutrient data from Copernicus Marine Environment Monitoring Service (CMEMS) reanalysis and current data from Simple Ocean Data Assimilation (SODA) reanalysis, we calculated the meridional $PO_4$ transport during the summers of El Niño's decaying phase (refer to "Data and Method" for details).

As the referee noted, both observational and model results confirm that the composite analysis of meridional $PO_4$ transport through the TS shows significantly positive signals across the entire TS region during El Niño decaying summers (Figure R1). We have incorporated Figure R1a into Figure 1d in the main text to further illustrate the primary nutrient source in the ECS region. In addition, the model results demonstrate that the correlation between runoff (r = 0.59**; Figure R2a) and SCHL is weaker compared to the correlation between meridional $PO_4$ transport through the TS (represented as the blue diagonal box in Figure R1b) and SCHL (r = 0.68**; Figure R2b). In addition to the TS region index, zonally-averaged meridional $PO_4$ transport flux in TS and SCHL correlations at 26.5°N (120.5°E–122.5°E) and 27.5°N (121.5°E–123.5°E) also exhibit significantly high positive correlations of 0.66 and 0.57, respectively (Figure R3). Moreover, the distinction between the strong blooming and non-blooming groups reveals a pronounced difference in the magnitude of meridional $PO_4$ transport through the TS (Figure R4). We have incorporated Figure R2b and Figure R4 as Figure 6 in the main text. Therefore, we have added a paragraph to the

main text to describe the TS-transport-driven blooming mechanism based on the model results:

(L256-265): We also investigated the TS-transport-driven blooming mechanism, which is well known for its significant role in nutrient supply through the TS. We analyzed the meridional PO4 transport by comparing two groups as the same as Fig. 5, and found remarkably distinct differences between them (Figure 6a-c). Notably, within the TS, a strong positive (negative) signal was identified in the SB (NB) group. For the SB group, the nutrient flux exhibits a significant positive signal reaching the target region, potentially supplying large amount of nutrients and making a substantial contribution to SCHL blooming in the ECS region. Furthermore, a positive correlation of 0.68—higher than the runoff-driven mechanism—was identified between the anomalous SCHL blooming magnitude in the target region and the meridional PO4 transport index indicated by the blue diagonal box in the TS region (Figure. 6d). This underscores the critical role of the TS-transport-driven mechanism in fostering nutrient into the ECS region.

[Figure]

**Figure R1**. Composite map of meridional PO₄ transport anomalies during El Niño decaying summer season in the Taiwan Strait (TS) for all El Niño cases in observations (Left Panel) and model results (Right Panel). All the black dots where the responses are statistically significant at the 90% (in observations) and 95% (in model results) confidence level, determined using the bootstrap method.

[Figure]

**Figure R2**. The relationship between area-averaged runoff and SCHL anomalies over the target region (Figure R2a), and the relationship between area-averaged meiridonal PO₄ transport and SCHL anomalies over the target region (Figure R2b).

[Figure]

**Figure R3**. The relationship between zonally-averaged meridional PO4 transport (26.5˚N-120.5˚E-122.5˚E; Left Panel / 27.5˚N – 121.5˚E-123.5˚E; Right Panel) and SCHL anomalies over the target region.

[Figure]

**Figure R4**. Composite maps of meridional PO₄ transport anomalies during El Niño decaying summer for Strong Blooming group (Figure R4a), Non-Blooming group (Figure R4b) and the difference between the two groups (Figure R4c).

The overall structure of the manuscript has been substantially reorganized. Instead of comparing the two previously proposed blooming mechanisms—"runoff-driven" and "upwelling-driven"—we have revised the manuscript to first re-evaluate the known mechanisms: "runoff-driven" and "TS-transport-driven". Subsequently, we incorporate both observational and modeling evidence to support the upwelling-driven mechanism, emphasizing the importance of understanding all these complex phytoplankton blooming mechanisms.

Additionally, in the revised manuscript, we assessed the relative contributions of three mechanisms using a regression method instead of the joint-composite analyses (Figure R5, Table R1; We added Figure R5 and Table R1 as Figure 9 and Table 1 in the manuscript respectively). We normalized all variables and conducted multiple regression for three mechanism indices at each grid point of SCHL anomalies across the target region. And, we quantified the relative contributions of three mechanisms by multiplying the linear regression coefficients between the Nino3.4 index and each mechanism index by their multiple regression coefficients respectively (Eq. 1; We added it as Eq. 4 in the manuscript).

The results indicate that both the Runoff- and the TS-transport-driven mechanism have the most significant contributions to the intensity of SCHL blooming in the target region. Although the Ekman Upwelling-driven mechanism exhibits a lower contribution compared to the others, it still exerts a meaningful influence on about half of the other mechanisms. Spatially, the Runoff-driven mechanism primarily affects the central area of the target region (Figure R5a), while the TS-transport-driven mechanism plays a major role along a southwest-northeast diagonally (Figure R5b). In contrast, the Ekman Upwelling-driven mechanism uniformly influences the overall target region (Figure R5c). Especially, the runoff- and the TS-transport-driven mechanism contribute similarly to the SCHL blooming intensity in the target region.

Although the Ekman upwelling-driven mechanism accounts for approximately 40 to 47% of the others, it still exerts a significant influence (Table R1). Consequently, we have replaced the paragraph that calculated the relative contribution for each mechanism from the joint-composite analyses previously we used to the regression method as follows:

(L313-349): So far, we have conducted a comprehensive evaluation and analysis of the complex mechanisms driving phytoplankton blooms in the ECS during the summers following the decaying phase of El Niño events. In addition to the runoff- and TS-transport-driven mechanisms suggested in previous studies, we introduced the Ekman upwelling-driven blooming mechanism. Subsequently, we quantified the relative contributions of these three mechanisms based on regression methods using (Eq. 4) as follows:

$$\frac{dChl}{dNino3.4}(\alpha) = \frac{\partial Chl}{\partial Runoff} \times \frac{dRunoff}{dNino3.4} + \frac{\partial Chl}{\partial TS-transport} \times \frac{dTS-transport}{dNino3.4} +$$

$$\frac{\partial Chl}{\partial Ekman-Upwelling} \times \frac{dEkman-Upwelling}{dNino3.4} + residual \qquad (4)$$

Firstly, we normalized all variables and conducted the multiple regression with respect to SCHL index in the target region as the dependent variable and three mechanism indices as independent variables. Secondly, we calculated each term by multiplying the linear regreesion coefficient between Nino3.4 index and each mechanism index by the corresponding multiple regression coefficient for each mechanism index. The combined effects of the three mechanisms and the residual term collectively account for the variation of the SCHL index in the ECS region with respect to the El Niño events. Therefore, each term indicates how changes in three mechanism induced by ENSO cycle (as indicated by the Nino3.4 index) affect on phytoplankton blooms in the ECS region. The results show the effect of runoff-driven mechanism and the TS-transport mechanisms to SCHL blooming intensity in the target region are comparable (Table 1). In the case of the Ekman upwelling mechanism, it accounts for about 40% to 47% of contributions comapred to the other two mechanisms, yet still exerts a meaningful influence as well.

**Table R1.** Relative contributions of three mechanisms to SCHL blooming in the target region.

| | $\alpha$ | Runoff | TS-transport | Ekman Upwelling |
|---|---|---|---|---|
| Contribution | 0.511 | 0.152 | 0.138 | 0.065 |

Additionally, we identified their spatial contributions to further elucidate the effects of each mechanism within the target region (Figure 9). We conducted multiple regression for three mechanism indices at each grid point of SCHL anomalies across the target region. Following the same way examined above, we calculated the spatial contributions by multiplying the linear regression coefficents of each mechanism with repsect to the Nino3.4 index. The results showed that the runoff-driven mechanism exhibited the strongest contribution at the center of the target region, driven by direct riverine nutrient inflow (though the model doesn't simulate P input) and nutrient supply through buoyancy upwelling induced by river discharge (Figure 9a). In contrast, the TS-transport-driven mechanism significantly contributes along a southwest-northeast diagonal direction, aligning with nutrient transport via the TS from the SCS (Figure 9b). Additionally, the newly proposed Ekman upwelling-driven mechanism, while relatively lower in overall contribution, shows significant and uniformly distributed impacts across the target region (Figure 9c). These results suggest that all mechanisms can affect comprehensively to induce phytoplankton blooms in the ECS during the following summers of the decaying phase of El Niño events. Therefore, considering all mechanisms is essential for accurately predicting the intensity of phytoplankton blooms.

[Figure]

**Figure R5.** Relative contributions of three mechanisms to SCHL blooming in the target region during summers following the decaying phase of El Niño events **(a)** Runoff-driven mechanism **(b)** TS-transport-driven mechanism **(c)** Ekman upwelling-driven mechanism.

We have revised the title of the manuscript to **"Phytoplankton Blooming Mechanisms over the East China Sea during El Niño Decaying Summers"** to better reflect the scope and findings of our study.

Additionally, we have annotated all correlation analyses to emphasize statistical significance: a star (*) was added next to r values with a p-value below 0.05, and double stars (**) was added for r values with a p-value below 0.01.

We believe that these substantial revisions comprehensively address the referee's concerns and enhance the scientific value of the manuscript.

**Reference**

Chen, C. A.: Buoyancy leads to high productivity of the Changjiang diluted water: a note, Acta Oceanol. Sin., 27, 133–140, 2008.

Chen, C. T. A.: The Three Gorges Dam: Reducing the upwelling and thus productivity in the East China Sea, Geophys. Res. Lett., 27, 381–383, https://doi.org/10.1029/1999GL002373, 2000.

Chen, C. T. A., Liu, C. T., Chuang, W. S., Yang, Y. J., Shiah, F. K., Tang, T. Y., and Chung, S. W.: Enhanced buoyancy and hence upwelling of subsurface Kuroshio waters after a typhoon in the southern East China Sea, J. Mar. Syst., 42, 65–79, https://doi.org/10.1016/S0924-7963(03)00065-4, 2003.

Chen, C. T. A., Yeh, Y. T., Chen, Y. C., and Huang, T. H.: Seasonal and ENSO-related interannual variability of subsurface fronts separating West Philippine Sea waters from South China Sea waters near the Luzon Strait, Deep. Res. Part I Oceanogr. Res. Pap., 103, 13–23, https://doi.org/10.1016/j.dsr.2015.05.002, 2015.

Hill, A. E.: Buoyancy effects in coastal and shelf seas, Sea, 21–62, 1998.

Huang, T. H., Chen, C. T. A., Zhang, W. Z., and Zhuang, X. F.: Varying intensity of Kuroshio intrusion into Southeast Taiwan Strait during ENSO events, Cont. Shelf Res., 103, 79–87, https://doi.org/10.1016/j.csr.2015.04.021, 2015.

Zhang, W., Zhuang, X., Chen, C. A., and Huang, T.: The impact of Kuroshio water on the source water of the southeastern Taiwan Strait: numerical results, Acta Oceanol. Sin., 34, 23–34, https://doi.org/10.1007/s13131-015-0720-x, 2015.

---

## Author Response (AR1)

**Review of "Two different phytoplankton blooming mechanisms over the East China Sea during El-Niño decaying summers" by Lee et al.**

**Comment on egusphere-2024-3406 | Anonymous Referee #1:**

This manuscript discusses an important issue of international interest. However, it is premature because of major flaws.

The major flaw is the erroneous assumption that riverine input is the primary source of nutrients(line 27). Much of the Introduction and Methods concerns river input. However, the Kuroshio Intermediate Water is the primary source of nutrients for the East China Sea shelf. Many papers substantiated this notion. The authors used the results presented in Figs. 2 and 3 to prove the importance of riverine inputs of nutrients. Yet, Figs. 2 and 3 only provided a good correlation between chlorophyll and nutrients that mostly did not come from rivers. Instead, the upwelling of nutrient-rich Kuroshio subsurface waters provided most of the nutrients. Of course, upwelling could be induced by the buoyance effect caused by the river water(e.g., Chen 2008, Acta Oceanologica Sinica, 27,133, 2008). To summarize, the so-called runoff-driven blooming mechanism is not caused by riverine nutrients. Instead, it is caused by buoyancy-driven upwelling and vertical mixing. BTW, validation is needed to substantiate the modeled results presented in these two figures.

Another factor worth mentioning is that the outflow of the South China Sea is the primary source of Kuroshio waters entering the East China Sea. During El Nino years, there is a more substantial outflow of the more nutrient-rich SCS water(e.g., Chen et al., Deep-Sea Res. I, 103,13, 2015), which may enhance productivity on the ECS shelf.

As a final note, many of the figures are not labeled correctly. Figures 1, 2, 3, 4, 5, 6, 9, and 10 are all related to anomalies, not the actual values. The "p" values should also be provided.

**Response:** We thank the reviewer for constructive and valuable comments and suggestions on our results. We have made substantial revisions to the manuscript in response to the referee's key comments.

We sincerely appreciate referee #1 for the comment that nutrient supply from the entrainment of subsurface waters via dynamic buoyancy upwelling induced by great out-flowing river plume is an important influence on phytoplankton blooms in the ECS region as well. Obviously, we missed this physical process in the original manuscript. By following the referee's suggestions, we have added a clarification to the manuscript to state that the runoff-driven mechanism also includes nutrient supply from the subsurface water entrainment due to buoyancy upwelling caused by enhanced river discharge during the El Niño decaying summers in the ECS continental shelves as follows:

(L139-145): In addition to direct riverine nutrient input, enhanced river discharge can deliver nutrients to the East China Sea (ECS) region through buoyancy-driven upwelling (Chen, 2008; Chen et al., 2003; Chen, 2000; Hill, 1998). When the outflowing river plume becomes more substantial than the incoming river discharge, subsurface waters are upwelled to maintain water balance. This process allows nutrient-rich Kuroshio subsurface waters to ascend along the ECS shelf edge, supplying essential nutrients to the region. Unlike wind-driven upwelling, this buoyancy-driven upwelling is governed by the physical properties of the subsurface waters in response to intensified river outflow.

(L170-174): Although the GFDL-CM2.1 ESM does not simulate P inputs from riverine inflows, it does account for P supply through the dynamic ascent of subsurface waters driven by buoyancy upwelling resulting from river discharge. Therefore, beyond analyses based on observational and reanalysis data, long-term model simulations enable us to comprehensively understand the phytoplankton blooming mechanisms in the ECS during the following summers of El Niño events.

Additionally, we have substantially revised the manuscript in response to the referee's comments. As the reviewer suggested, we discussed the role of nutrient transport through the Taiwan Strait (TS) (Huang et al., 2015; Zhang et al., 2015; Chen et al., 2015) by adding new analyses. In the revised manuscript, we have extensively addressed this by incorporating the nutrient transport mechanism through the TS (TS-transport-driven mechanism). Using nutrient data from Copernicus Marine Environment Monitoring Service (CMEMS) reanalysis and current data from Simple Ocean Data Assimilation (SODA) reanalysis, we calculated the meridional $PO_4$ transport during the summers of El Niño's decaying phase (refer to "Data and Method" for details).

As the referee noted, both observational and model results confirm that the composite analysis of meridional $PO_4$ transport through the TS shows significantly positive signals across the entire TS region during El Niño decaying summers (Figure R1-1). We have incorporated Figure R1-1a into Figure 1d in the main text to further illustrate the primary nutrient source in the ECS region. In addition, the model results demonstrate that the correlation between runoff (r = 0.59**; Figure R1-2a) and SCHL is weaker compared to the correlation between meridional $PO_4$ transport through the TS (represented as the blue diagonal box in Figure R1-1b) and SCHL (r = 0.68**; Figure R1-2b). In addition to the TS region index, zonally-averaged meridional $PO_4$ transport flux in TS and SCHL correlations at 26.5°N (120.5°E–122.5°E) and 27.5°N (121.5°E–123.5°E) also exhibit significantly high positive correlations of 0.66 and 0.57, respectively (Figure R1-3). Moreover, the distinction between the strong blooming and non-blooming groups reveals a pronounced difference in the magnitude of meridional $PO_4$ transport through the TS (Figure R1-4). We have incorporated Figure R1-2b and Figure R1-4 as Figure 6 in the main text. Therefore, we have added a paragraph to

the main text to describe the TS-transport-driven blooming mechanism based on the model results:

(L256-265): We also investigated the TS-transport-driven blooming mechanism, which is well known for its significant role in nutrient supply through the TS. We analyzed the meridional PO4 transport by comparing two groups as the same as Fig. 5, and found remarkably distinct differences between them (Figure 6a-c). Notably, within the TS, a strong positive (negative) signal was identified in the SB (NB) group. For the SB group, the nutrient flux exhibits a significant positive signal reaching the target region, potentially supplying large amount of nutrients and making a substantial contribution to SCHL blooming in the ECS region. Furthermore, a positive correlation of 0.68—higher than the runoff-driven mechanism—was identified between the anomalous SCHL blooming magnitude in the target region and the meridional PO4 transport index indicated by the blue diagonal box in the TS region (Figure. 6d). This underscores the critical role of the TS-transport-driven mechanism in fostering nutrient into the ECS region.

[Figure]

**Figure R1-1**. Composite map of meridional PO$_4$ transport anomalies during El Niño decaying summer season in the Taiwan Strait (TS) for all El Niño cases in observations (Left Panel) and model results (Right Panel). All the black dots where the responses are statistically significant at the 90% (in observations) and 95% (in model results) confidence level, determined using the bootstrap method.

[Figure]

**Figure R1-2**. The relationship between area-averaged runoff and SCHL anomalies over the target region (Figure R1-2a), and the relationship between area-averaged meridional $PO_4$ transport and SCHL anomalies over the target region (Figure R1-2b).

[Figure]

**Figure R1-3**. The relationship between zonally-averaged meridional PO4 transport (26.5°N-120.5°E-122.5°E; Left Panel / 27.5°N – 121.5°E-123.5°E; Right Panel) and SCHL anomalies over the target region.

[Figure]

**Figure R1-4**. Composite maps of meridional $PO_4$ transport anomalies during El Niño decaying summer for Strong Blooming group (Figure R4a), Non-Blooming group (Figure R1-4b) and the difference between the two groups (Figure R1-4c).

The overall structure of the manuscript has been substantially reorganized. Instead of comparing the two previously proposed blooming mechanisms—"runoff-driven" and "upwelling-driven"—we have revised the manuscript to first re-evaluate the known mechanisms: "runoff-driven" and "TS-transport-driven". Subsequently, we incorporate both observational and modeling evidence to support the upwelling-driven mechanism, emphasizing the importance of understanding all these complex phytoplankton blooming mechanisms.

Additionally, in the revised manuscript, we assessed the relative contributions of three mechanisms using a regression method instead of the joint-composite analyses (Figure R1-5, Table R1-1; We added Figure R1-5 and Table R1-1 as Figure 9 and Table 1 in the manuscript respectively). We normalized all variables and conducted multiple regression for three mechanism indices at each grid point of SCHL anomalies across the target region. And, we quantified the relative contributions of three mechanisms by multiplying the linear regression coefficients between the Nino3.4 index and each mechanism index by their multiple regression coefficients respectively (Eq. R1-1; We added it as Eq. 4 in the manuscript).

$$\frac{dChl}{dNino3.4}(\alpha) = \frac{\partial Chl}{\partial Runoff} \times \frac{dRunoff}{dNino3.4} + \frac{\partial Chl}{\partial TS-transport} \times \frac{dTS-transport}{dNino3.4} +$$

$$\frac{\partial Chl}{\partial Ekman-Upwelling} \times \frac{dEkman-Upwelling}{dNino3.4} + residual \qquad \text{(Eq. R1-1)}$$

The results indicate that both the Runoff- and the TS-transport-driven mechanism have the most significant contributions to the intensity of SCHL blooming in the target region. Although the Ekman Upwelling-driven mechanism exhibits a lower

contribution compared to the others, it still exerts a meaningful influence on about half of the other mechanisms. Spatially, the Runoff-driven mechanism primarily affects the central area of the target region (Figure R1-5a), while the TS-transport-driven mechanism plays a major role along a southwest-northeast diagonally (Figure R1-5b). In contrast, the Ekman Upwelling-driven mechanism uniformly influences the overall target region (Figure R1-5c). Especially, the runoff- and the TS-transport-driven mechanism contribute similarly to the SCHL blooming intensity in the target region. Although the Ekman upwelling-driven mechanism accounts for approximately 40 to 47% of the others, it still exerts a significant influence (Table R1-1). Consequently, we have replaced the paragraph that calculated the relative contribution for each mechanism from the joint-composite analyses previously we used to the regression method as follows:

(L313-349): So far, we have conducted a comprehensive evaluation and analysis of the complex mechanisms driving phytoplankton blooms in the ECS during the summers following the decaying phase of El Niño events. In addition to the runoff- and TS-transport-driven mechanisms suggested in previous studies, we introduced the Ekman upwelling-driven blooming mechanism. Subsequently, we quantified the relative contributions of these three mechanisms based on regression methods using (Eq. 4) as follows:

$$\frac{dChl}{dNino3.4}(\alpha) = \frac{\partial Chl}{\partial Runoff} \times \frac{dRunoff}{dNino3.4} + \frac{\partial Chl}{\partial TS-transport} \times \frac{dTS-transport}{dNino3.4} +$$

$$\frac{\partial Chl}{\partial Ekman-Upwelling} \times \frac{dEkman-Upwelling}{dNino3.4} + residual \qquad (4)$$

Firstly, we normalized all variables and conducted the multiple regression with respect to SCHL index in the target region as the dependent variable and three mechanism indices as independent variables. Secondly, we calculated each term by multiplying the linear regreesion coefficient between Nino3.4 index and each mechanism index by the corresponding multiple regression coefficient for each mechanism index. The combined effects of the three

mechanisms and the residual term collectively account for the variation of the SCHL index in the ECS region with respect to the El Niño events. Therefore, each term indicates how changes in three mechanism induced by ENSO cycle (as indicated by the Nino3.4 index) affect on phytoplankton blooms in the ECS region. The results show the effect of runoff-driven mechanism and the TS-transport mechanisms to SCHL blooming intensity in the target region are comparable (Table 1). In the case of the Ekman upwelling mechanism, it accounts for about 40% to 47% of contributions comapred to the other two mechanisms, yet still exerts a meaningful influence as well.

**Table R1-1.** Relative contributions of three mechanisms to SCHL blooming in the target region.

|  | $\alpha$ | Runoff | TS-transport | Ekman Upwelling |
|---|---|---|---|---|
| Contribution | 0.511 | 0.152 | 0.138 | 0.065 |

Additionally, we identified their spatial contributions to further elucidate the effects of each mechanism within the target region (Figure 9). We conducted multiple regression for three mechanism indices at each grid point of SCHL anomalies across the target region. Following the same way examined above, we calculated the spatial contributions by multiplying the linear regression coeffcients of each mechanism with repsect to the Nino3.4 index. The results showed that the runoff-driven mechanism exhibited the strongest contribution at the center of the target region, driven by direct riverine nutrient inflow (though the model doesn't simulate P input) and nutrient supply through buoyancy upwelling induced by river discharge (Figure 9a). In contrast, the TS-transport-driven mechanism significantly contributes along a southwest-northeast diagonal direction, aligning with nutrient transport via the TS from the SCS (Figure 9b). Additionally, the newly proposed Ekman upwelling-driven mechanism, while relatively lower in overall contribution, shows significant and uniformly distributed impacts across the target region (Figure 9c). These results suggest that all mechanisms can affect comprehensively to induce phytoplankton blooms in the ECS during the following summers of the decaying phase of El Niño events. Therefore, considering all mechanisms is essential for accurately predicting the intensity of phytoplankton blooms.

[Figure]

**Figure R1-5.** Relative contributions of three mechanisms to SCHL blooming in the target region during summers following the decaying phase of El Niño events **(a)** Runoff-driven mechanism **(b)** TS-transport-driven mechanism **(c)** Ekman upwelling-driven mechanism.

We have revised the title of the manuscript to **"Phytoplankton Blooming Mechanisms over the East China Sea during El Niño Decaying Summers"** to better reflect the scope and findings of our study.

Additionally, we have annotated all correlation analyses to emphasize statistical significance: a star (*) was added next to r values with a p-value below 0.05, and double stars (**) was added for r values with a p-value below 0.01.

We believe that these substantial revisions comprehensively address the referee's concerns and enhance the scientific value of the manuscript.

**Comment on egusphere-2024-3406 | Anonymous Referee #2:**

Based on limited observations, previous studies suggested that in post-El Nino summers (JJA), increased runoff from the Yangtze River reduces salinity and increases Ch-a concentrations in the surface layer of the East China Sea (ECS). The present study extends the previous studies by using longer observations, suggesting that the western North Pacific anticyclone (WNPAC) in the lower troposphere during post-El Nino summers contributes to positive Ch-a anomalies in the ECS also through positive wind stress curl anomalies, which enhance upwelling of high nutrients from below the seasonal thermocline. The paper is generally well written and should be eventually published after addressing the following concerns.

**Response:** We thank referee #2 for encouraging and valuable comments on our manuscript. We have thoroughly addressed the feedback and revised the manuscript by fully incorporating the referee's suggestions.

**Major comments.**

1. The new mechanism of wind curl-induced upwelling is mostly based on a coarse-resolution ESM but the observational support is marginal. While observed Ch-a increase during post-El Nino summers is east of the Yangtze River estuary (Fig. 1a), positive wind curl anomalies are further to the south (Fig. 9b), almost entirely outside the ECS. The geographical discrepancy between Ch-a and wind curl anomalies needs to be reconciled. Just a thought: rainfall over eastern China takes a while to reach the coast and affects ECS Ch-a. Could this delay be important? Also there are many big reservoirs along the Yangtze River and water is being diverted from the river. Are these human

controls important for discharges at the estuary?

**Response:** We indeed thank the referee's valuable comments and suggestions. We have thoroughly revised the manuscript, completely reorganizing and rewriting it to emphasize the significance of our findings.

We have newly incorporated the contribution of nutrient supply through the TS to phytoplankton blooming in the ECS into the revised manuscript, as supported by several references (Huang et al., 2015; Chen et al., 2015; Zhang et al., 2015). Both observational data and model results reveal that nutrient transport through the TS into the ECS is intensified during El Niño decaying summers (Figure R2-1; we added Figure R2-1a as Figure 1d to the revised manuscript). This enhancement is driven by the intensification of southwesterly winds over the South China Sea (SCS), which is facilitated by the development of the Western North Pacific Anticyclone (WNPAC) during the El Niño decaying summers. We have added the following paragraph to the revised manuscript to describe the TS-transport-driven blooming mechanism based on the model results:

(L256-265): We also investigated the TS-transport-driven blooming mechanism, which is well known for its significant role in nutrient supply through the TS. We analyzed the meridional PO4 transport by comparing two groups as the same as Fig. 5, and found remarkably distinct differences between them (Figure 6a-c). Notably, within the TS, a strong positive (negative) signal was identified in the SB (NB) group. For the SB group, the nutrient flux exhibits a significant positive signal reaching the target region, potentially supplying large amount of nutrients and making a substantial contribution to SCHL blooming in the ECS region. Furthermore, a positive correlation of 0.68—higher than the runoff-driven mechanism—was identified between the anomalous SCHL blooming magnitude in the target region and the meridional PO4 transport index indicated by the blue diagonal box in the TS region (Figure.

6d). This underscores the critical role of the TS-transport-driven mechanism in fostering nutrient into the ECS region.

[Figure]

**Figure R2-1**. Composite map of meridional $PO_4$ transport anomalies during El Niño decaying summer season in the Taiwan Strait (TS) for all El Niño cases in observations (Left Panel) and model results (Right Panel). All the black dots where the responses are statistically significant at the 90% (in observations) and 95% (in model results) confidence level, determined using the bootstrap method.

Furthermore, our model results also showed that meridional phosphate ($PO_4$) transport through the TS contributes more significantly to SCHL in the ECS compared to runoff-driven mechanism (Figure R2-2; we added the Figure R2-2b as Figure 6d to

the revised manuscript). Additionally, in the Strong Blooming (SB) group, the meridional PO4 transport through the TS was stronger (Figure R2-3; we added the Figure R2-3 to the revised manuscript as Figures 6a–c).

[Figure]

**Figure R2-2**. The relationship between area-averaged runoff and SCHL anomalies over the target region (Figure R2-2a), and the relationship between area-averaged meridional PO4 transport and SCHL anomalies over the target region (Figure R2-2b).

[Figure]

**Figure R2-3**. Composite maps of meridional PO4 transport anomalies during El Niño decaying summer for Strong Blooming group (Figure R2-3a), Non-Blooming group (Figure R2-3b) and the difference between the two groups (Figure R2-3c).

The ESM we employed is indeed a low-resolution model with a 1˚ grid. We also agree with the reviewer that the high-resolution model can reflect more realistic contributions of the various phenomena. However, since the wind stress curl pattern exhibits a somewhat large-scale pattern, we believe the coarse resolution model is capable of simulating our main mechanisms. Satellite observations indicate that anomalous SCHL blooming regions in the East China Sea (ECS) are strongly concentrated around the Yangtze River Estuary (YRE) during El Niño decaying

[Figure]

**Figure R2-4.** Climatological oceanic current over the East Asian Marginal Seas. Vectors indicate the oceanic current and shading shows the oceanic current speed. The red box indicates the Wind Stress Curl index as selected in the manuscript.

summers, with weaker diagonal signals extending northeast from Taiwan to the Korea Strait. The observed positive wind stress curl anomalies are widespread, covering areas from the YRE through the Taiwan Strait and extending to the Luzon Strait. In the East Asian Marginal Seas, including the ECS, are featured by a northward current from the TS and Kuroshio Current. Therefore, the climatological northward flow of the Kuroshio Current and the currents passing through the TS during summer enable the transport of anomalous upwelled nutrients due to the widespread cyclonic wind stress curl across the ECS and even to the Korea Strait (Figure R2-4).

Additionally, in the revised manuscript, we assessed the relative contributions of three mechanisms using a regression method instead of the joint-composite analyses (Figure R2-5, Table R1; We added Figure R2-5 and Table R2-1 as Figure 9 and Table 1 in the manuscript respectively). We normalized all variables and conducted multiple regression for three mechanism indices at each grid point of SCHL anomalies across the target region. And, we quantified the relative contributions of three mechanisms by multiplying the linear regression coefficients between the Nino3.4 index and each mechanism index by their multiple regression coefficients respectively (Eq. R2-1; We added it as Eq. 4 in the manuscript).

The results indicate that both the Runoff- and the TS-transport-driven mechanism have the most significant contributions to the intensity of SCHL blooming in the target region. Although the Ekman Upwelling-driven mechanism exhibits a lower contribution compared to the others, it still exerts a meaningful influence on about half of the other mechanisms. Spatially, the Runoff-driven mechanism primarily affects the central area of the target region (Figure R2-5a), while the TS-transport-driven mechanism plays a major role along a southwest-northeast diagonally (Figure R2-5b). In contrast, the Ekman Upwelling-driven mechanism uniformly influences the overall target region (Figure R2-5c). Especially, the runoff- and the TS-transport-driven mechanism contribute similarly to the SCHL blooming intensity in the target region. Although the Ekman upwelling-driven mechanism accounts for approximately 40 to 47% of the others, it still exerts a significant influence (Table R2-1). Consequently, we have replaced the paragraph that calculated the relative contribution for each mechanism from the joint-composite analyses previously we used to the regression method as follows:

[revised manuscript text omitted]

Our modeling analyses suggest that the upwelling-driven mechanism operates as a supplemental mechanism, therefore, even if the location of maximum upwelling does not always coincide with areas of maximum phytoplankton blooms.

In addition. we investigated the lagged relationship between summer SCHL in the target region and monthly precipitation over the eastern China to analyze the time delay of the effect of rainfall on the SCHL. Our results indicate that summer season SCHL in the ECS region is most strongly correlated with concurrent summer precipitation (Figure R2-6). In particular, the July–August precipitation in the observations and the June–July precipitation in the model each showed the highest correlation with summer SCHL. Moreover, for both the observations and the model, summer SCHL exhibited the highest correlation with JJA seasonal precipitation. Therefore, we believe that time delay in the impact of rainfall on SCHL in the ECS region does not significantly compromise the validity of our study.

[Figure]

**Figure R2-6.** The plot shows the lagged relationship between precipitation during summer season (June-July-August; JJA) and the surface Chlorophyll-a (SCHL) in the target region of both **(a-b)** observations and **(c-d)** model. **(a)** The plot shows 6 month lagged relationship between summer season SCHL and monthly precipitation in the eastern China (Observations: 30.5˚N-32.5˚N / 114˚E-119˚E; Model: 27.5˚N-33.5˚N / 117.5˚E-124.5˚E, which is broader than observations) from June (0) to the November (0). **(b)** The plot shows the lagged relationship between the averaged summer season SCHL and the 3-month averaged precipitation from JJA (0) to SON (0). **(c-d)** As same as Fig. R6a-b, but the case of model results. The deeper red colors indicate that the p-value is below 0.1 (0.05), signifying statistical significance at the 90% (95%) confidence level for observations (model).

Indeed, in terms of anthropogenic impacts, the three Gorges Dam (TGD), located on the Yangtze River, has a substantial impact on the volume of river discharge flowing into the ECS. Increased water storage of the TGD leads to a reduction in river discharge into the ECS, which in turn causes ecological changes such as shifts in chlorophyll-a and other marine ecosystem components (Jiao et al., 2007).

Furthermore, we have added the description that the runoff-driven mechanism encompasses nutrient input resulting from the entrainment of subsurface waters via buoyancy upwelling by following another referee's comment as follows:

(L139-145): In addition to the direct riverine nutrient input, enhanced river discharge can facilitate nutrients to the ECS region through buoyancy upwelling (Chen, 2008; Chen et al., 2003; Chen, 2000; Hill, 1998). When the out-flowing river plume becomes more substantial than the incoming river discharge, subsurface waters are readily upwelled to preserve the water balance. Consequently, the nutrient-rich Kuroshio subsurface waters ascend along the ECS shelf edge, providing essential nutrients. This buoyancy-driven upwelling occurs independently of wind conditions, driven primarily by the physical properties of the subsurface waters in response to the intensified river outflow.

(L170-174): Although the GFDL-CM2.1 ESM does not simulate P inputs from riverine inflows, it does account for P supply through the dynamic ascent of subsurface waters driven by buoyancy upwelling resulting from river discharge. Therefore, beyond analyses based on observational and reanalysis data, long-term model simulations enable us to comprehensively understand the phytoplankton blooming mechanisms in the ECS during the following summers of El Niño events.

We also have revised the title of the manuscript to **"Phytoplankton Blooming Mechanisms over the East China Sea during El Niño Decaying Summers"** to better reflect the scope and findings of our study.

Therefore, the extensively revised manuscript offers a more comprehensive understanding of the mechanisms driving phytoplankton blooming in the ECS during El Niño decaying summers, improving the previous manuscript.

2. ENSO is but one driver, but the WNPAC is an intrinsic mode of Asian summer monsoon variability and could be active in non-ENSO summers (P. Zhang et al. 2024, *JC*). The super-active Meiyu season in 2020 is a recent example (Z.Q. Zhou et al. 2021, *PNAS*). Could this explain the discrepancy between Figs. 1a and 9b?

**Response:** We appreciate the referee's suggestion regarding the mechanisms behind anomalous SCHL blooming that occur during Non-ENSO summers. Indeed, as the 2020 Super-active Meiyu season took place without considerable ENSO events (Figure R2-7). Specifically, the anomalous SCHL bloom intensity in 2020 (0.124 mg m$^-$$^3$) is weaker than the average bloom intensity during El Niño periods (0.174 mg m$^{-3}$), despite experiencing the highest precipitation since 1961 and the consequent substantial river discharge (Zhou et al., 2021).

This finding indicates that SCHL blooming in the ECS is likely driven by a combination of several mechanisms. In addition to enhanced river discharge, nutrient transport through the TS and the supply of upwelled nutrients from sub-surface layer to the ECS via ocean currents—facilitated by a broad positive wind stress curl

extending from the Yangtze River to the Luzon Strait—also play significant roles during the decaying summer season of El Niño.

Therefore, our study reveals that the upwelling-driven mechanism can play an additional role in phytoplankton blooming within the ECS with other mechanisms (runoff-driven and TS-transport-driven). The contributions of the runoff-driven and TS-transport-driven mechanisms to blooming are greater, thus, there may be a spatial discrepancy between the regions exhibiting the most intense anomalous surface chlorophyll-a (SCHL) blooms and the areas where upwelling occurs.

[Figure]

**Figure R2-7**. As same as the Figure 1b in revised manuscript, and the red circle indicates the case of 2020 supter-active Meiyu summer season.

**Minor comments**

1. The discussion of WNPAC dynamics is quite dated (L235-240, L386-394). See

Xie et al. (2016, Adv Atmos Sci) and Chowdary et al. (2019, Current Clim Change Reps) for recent reviews.

**Response:** Thanks for suggesting the papers. We cited the references in the revised manuscript.

2. Regarding the global warming effect on WNPAC (L383-386), a CMIP6 analysis suggests that ENSO-induced variability does not change much but ENSO-unrelated variability intensifies with warming (C.Y. Wang et al. 2023, JC). This would imply a weakened ENSO effect on ECS Ch-a.

**Response:** We have incorporated the references in the revised manuscript.

3. The writing is overall good but please check English grammar. L21: add "the" in front of "East China Sea". L35: add "The" in front of "Western North."

**Response:** Corrected.

**Comment on egusphere-2024-3406 | Anonymous Referee #3:**

**General Comments:**

The study uses satellite remote sensing, reanalysis data, and model simulation to investigate two mechanisms (river discharge and upwelling) behind how ENSO cycle influences phytoplankton biomass in East China Sea, an area of fisheries and biogeochemical importances, nd is subjected to anthropogenic perturbations.

**Response:** We appreciate reviewer #3's helpful comments and valuable suggestions. We have revised the manuscript by fully incorporating the reviewer's comments. Our responses to the specific comments are as follows:

• The authors conclude that both mechanisms are relevant to the positive surface Chl a anomalies during ENSO phase based mainly on correlation analysis. However, correlation does not mean causality. If you rely on correlation analysis, the correlation between nutrient fluxes (horizontal transport or vertical transport) and Chl a under variable phases of ENSO cycle may be more robust.

**Response:** We appreciate Referee#3's suggestion. We have newly incorporated the contribution of nutrient supply through the Taiwan Strait (TS) to the anomalous increases in Surface Chlorophyll-a (SCHL) concentrations in the East China Sea (ECS) during the summer following the El Niño's events (Chen et al., 2015; Huang et al., 2015; Zhang et al., 2015). In the revised manuscript, we have extensively addressed this by incorporating the nutrient transport mechanism through the TS (TS-transport-driven mechanism). Using nutrient data from CMEMS reanalysis and oceanic current

data from Simple Ocean Data Assimilation (SODA) reanalysis, we calculated the meridional phosphate ($PO_4$) transport during the summers of El Niño's decaying phase (refer to "Data and Method" for details). Our findings reveal a markedly strong $PO_4$ transport through the TS, highlighting its critical role in driving phytoplankton blooms in the ECS (Figure R3-1, we added Figure R3-1a as Figure 1d to the revised manuscript).

[Figure]

**Figure R3-1**. Composite map of meridional $PO_4$ transport anomalies during El Niño decaying summer season in the Taiwan Strait (TS) for all El Niño cases in observations (Left Panel) and model results (Right Panel). All the black dots where the responses are statistically significant at the 90% (in obseravtions) and 95% (in model results) confidence level, determined using the bootstrap method.

In addition, the model results demonstrate that the correlation between runoff (r = 0.59\*\*; Figure R3-2a) and SCHL is weaker compared to the correlation between meridional $PO_4$ transport through the TS (represented as the blue diagonal box in Figure R3-1b) and SCHL (r = 0.68\*\*; Figure R3-2b). In addition to the TS region, the

meridional PO₄ transport flux in longitudinal line and SCHL correlations at 26.5°N (120.5°E–122.5°E) and 27.5°N (121.5°E–123.5°E) also exhibit significantly high positive correlations of 0.66 and 0.57, respectively (Figure R3-3). Moreover, the distinction between the strong blooming and non-blooming groups reveals a pronounced difference in the magnitude of meridional PO₄ transport through the TS (Figure R3-4). We have incorporated Figure R3-2b and Figure R3-4 as Figure 6 in the main text. Therefore, we have added a paragraph to the main text to describe the TS-transport-driven blooming mechanism based on the model results:

(L256-265): We also investigated the TS-transport-driven blooming mechanism, which is well known for its significant role in nutrient supply through the TS. We analyzed the meridional PO₄ transport by comparing two groups as the same as Figure 5, and found remarkably distinct differences between them (Figure 6a-c). Notably, within the TS, a strong positive (negative) signal was identified in the SB (NB) group. For the SB group, the nutrient flux exhibited a significant positive signal reaching the target region, potentially supplying large amount of nutrient and making a substantial contribution to SCHL blooming in the ECS region.

[Figure]

**Figure R3-2**. **(a)** The relationship between area-averaged runoff and SCHL anomalies over the target region, and **(b)** the relationship between area-averaged meridional PO₄ transport and SCHL anomalies over the target region.

Furthermore, a positive correlation of 0.68—higher than the runoff-driven mechanism—was identified between the anomalous SCHL blooming magnitude in the target region and meridional $PO_4$ transport index indicated by the blue diagonal box in the TS region (Figure 6d). This underscores the critical role of the TS-transport-driven mechanism in fostering nutrient into the ECS region.

[Figure]

**Figure R3-3**. The relationship between zonally-averaged meridional PO4 transport (26.5°N-120.5°E-122.5°E; Left Panel / 27.5°N – 121.5°E-123.5°E; Right Panel) and SCHL anomalies over the target region.

[Figure]

**Figure R3-4**. Composite maps of meridional $PO_4$ transport anomalies during El Niño decaying summer for **(a)** Strong Blooming group, **(b)** Non-Blooming group and the **(c)** difference between the two groups.

In response to the reviewer's comment, we calculated a monthly correlation of meridional $PO_4$ transport along a zonally averaged longitudinal line near the TS. This approach was employed to more robustly evaluate the influence of nutrient flux

through the TS on SCHL over the ECS region during the variable ENSO cycle. Both observational data and model results reveal that the correlation is strongest during the summer season (Figure R3-5). This seasonal analysis underscores the significant role of PO$_4$ transport through the TS in influencing phytoplankton blooms in the ECS especially during the summer season.

[Figure]

**Figure R3-5**. The plots indicate the seasonal variation of correlation coefficient between surface chlorophyll-a (SCHL) during summer season (June-July-August; JJA) in the target region and monthly meridional PO$_4$ transport in Taiwan Strait (TS) by zonally-averaged at specific latitude (OBS : 25.75˚N / 119.5˚E-124˚E; Model : 26.5˚N / 120.5˚E-122.5˚E) in both **(a)** reanalysis data and **(b)** model. The deeper red colors indicate that the p-value is below 0.05, signifying statistical significance at the 95% confidence level.

We also have revised the title of the manuscript to **"Phytoplankton Blooming Mechanisms over the East China Sea during El Niño Decaying Summers"** to better reflect the scope and findings of our study.

Furthermore, we have added the description that the runoff-driven mechanism encompasses nutrient input resulting from the entrainment of subsurface waters via buoyancy upwelling by following another referee's comment as follows:

(L139-145): In addition to the direct riverine nutrient input, enhanced river discharge can facilitate nutrients to the ECS region through buoyancy upwelling (Chen, 2008; Chen et al., 2003; Chen, 2000; Hill, 1998). When the out-flowing river plume becomes more substantial than the incoming river discharge, subsurface waters are readily upwelled to preserve the water balance. Consequently, the nutrient-rich Kuroshio subsurface waters ascend along the ECS shelf edge, providing essential nutrients. This buoyancy-driven upwelling occurs independently of wind conditions, driven primarily by the physical properties of the subsurface waters in response to the intensified river outflow.

(L170-174): Although the GFDL-CM2.1 ESM does not simulate P inputs from riverine inflows, it does account for P supply through the dynamic ascent of subsurface waters driven by buoyancy upwelling resulting from river discharge. Therefore, beyond analyses based on observational and reanalysis data, long-term model simulations enable us to comprehensively understand the phytoplankton blooming mechanisms in the ECS during the following summers of El Niño events.

Therefore, we believe the extensively revised manuscript offers a more comprehensive understanding of the mechanisms driving phytoplankton blooming in the ECS during El Niño decaying summers.

• The main weakness of this study is the coarse resolution of the model used. With 1° by 1° resolution, mesoscale processes related to river discharge may not be resolved. In fact, the model failed to reproduce the distribution of Chl anomalies of ECS as observed in remote sensing (compare Figure 1a and Figure 3a). The modeled Chl distribution pattern is more related to Taiwan Strait Current from the south than due to river plume. Does the model sufficiently simulate the observed nutrient distributions?

**Response:** We thank for the referee's valuable comments about the reliability of the model simulations.

We agree with the reviewer that the Earth System Model (ESM) used in our study has a relatively coarse resolution of 1° × 1°. However, our analyses indicate that the mechanisms reported in observational studies are sufficiently replicated in the model results. Therefore, we believe that the coarse resolution model is also somewhat capable of simulating the mechanisms elucidated in our study.

Furthermore, satellite data tend to overestimate SCHL values near river estuary due to turbidity, as reported by Yamaguchi *et al* (2012). In addition, the GFDL-CM2.1 model used in our study does not simulate phosphate inflow from river discharge, which can lead to an underestimation of SCHL anomalies compared to satellite data.

Dunne *et al* 2013 showed the bias of biogeochemical variables ($NO_3$, $PO_4$, Chlorophyll-a) using both GFDL-ESM2M (z-coordinate model) and GFDL-ESM2G (isopycnal-coordinate model) models compared to observations. For nutrients ($NO_3$, $PO_4$), both models exhibited a high spatial variance captured coefficient ($r^2 > 0.8$), indicating effective replication of global spatial patterns. In the ECS, the model underestimated nitrate ($NO_3$) input from river runoff, resulting in an underestimation of

nutrient concentrations near the Yangtze River estuary (YRE; Figure R3-6a). Model simulation is improved as the distance from the estuary increases. The model does not simulate the riverine input of PO4, which leads to a slight underestimation of $PO_4$ concentrations from the YRE to the Bohai Sea (Figure R3-6b). Similar to $NO_3$, the model's accuracy for $PO_4$ improved notably in offshore areas.

However, our model successfully replicates phytoplankton blooms in the ECS region, driven by nutrient inputs from river discharge, and additional nutrient transport through the TS as reported in observational studies. Furthermore, we identified that both runoff- and TS-transport-driven mechanisms are the primary contributors and confirmed that their spatial contributions are well reproduced, aligning with previous studies. Therefore, we believe that the model's resolution does not significantly impact the validity of our study.

[Figure]

**Figure R3-6**. Annual bias between observed and modeled **(a)** $NO_3$ and **(b)** $PO_4$ concentrations.

• The authors stated that phosphate input from river was not considered in the model, however within the Yangtze river estuary, there is a positive correlation between P and Chl (Figure 2c), that is, the Chl is stimulated by other water masses rich in phosphate. This can't be explained by increasing river water discharge, as Yangtze river water is high in N/P ratio in reality, and not included in the model. There are reports showing phosphate rich water with low N/P ratio is originated from Taiwan Strait Current (Huang et al., 2019). Therefore, the model might be simulating the straightening Taiwan Strait Current during ENSO phase, assuming the model represents the N/P correctly.

**Response:** We thank to the referee for the valuable comments regarding the nutrient sources in the ECS region.

As we responded above, we have substantially re-organized the manuscript. Initially, our study focused on the runoff-driven mechanism and the additional role of the upwelling-driven mechanism. However, we have now revised this manuscript to state the combined roles of the runoff-driven and TS-transport-driven mechanisms, with suggesting the additional role of upwelling-driven mechanism. This substantial reorganization enhances the clarity and value of our study, providing a more clear understanding of the mechanism influencing phytoplankton blooms in the ECS region.

Another caveat of simulating the Chl using plankton ecosystem model coupled with ocean GCM is that top-down control is not considered in majority of the models. Under same nutrient supply, Chl a may increase if grazers are suppressed by grazing of upper trophic levels.

As noted by the referee, the TOPAZ ecosystem model simulates the grazing rate by upper-level predators (Dunne et al., 2010). In this model, phytoplankton is detailed into three categories: large phytoplankton, small phytoplankton, and diazotroph phytoplankton. The grazing terms for small phytoplankton and diazotroph phytoplankton are simulated to increase proportionally to the 2nd power of their concentrations, while for large phytoplankton, the grazing rate increases proportionally to the 4/3 power of their concentration. Therefore, the grazing term driven by upper-level predators is inherently proportional to the chlorophyll-a concentration.

• The descriptions in the Data and Methods are in sufficient. It is not quite clear to me what the exact time frame of the analysis in this study. Line 65-66 stated 25 years of remote sensing data were used, does that mean the same time period is covered in model and re-analysis data? But in Figure 3, the figure legend mentioned 176 years of El Nino cases in ESM were analyzed. Some statistical analysis such as "joint composition analysis" may not be familiar to readers, me included, a reference should be very helpful.

**Response:** In our study, we utilized observational and reanalysis data spanning a 25-year period (1998–2022). To compensate for the relatively short duration and limitation of the datasets, we analyzed long-term simulation (1,000 years) under the present-day climate condition using GFDL-CM2.1. Using these 1,000-year simulations, we identified 176 El Niño cases, enabling a comprehensive analysis of the mechanisms driving chlorophyll blooming in the East China Sea during the El Niño decaying summer season using diverse El Niño cases. To provide additional clarification regarding the available period of model data, we have added the description to the "Data and Method" section as follows:

(L94-97): We used the present climate-based (1990-year atmospheric $CO_2$ concentration level; 353 parts per million (ppm)) long-term (1,000 years) simulations of the Geophysical Fluid Dynamic Laboratory (GFDL) - CM2.1 Earth System Model (ESM) fully coupled with the marine ecosystem model TOPAZv2 (Tracers of Ocean Phytoplankton with Allometric Zooplankton Version 2; Dunne et al., 2013).

Additionally, in the revised manuscript, we assessed the relative contributions of three mechanisms using a regression method instead of the joint-composite analyses (Figure R3-7, Table R3-1; We added Figure R3-7 and Table R3-1 as Figure 9 and Table 1 in the manuscript respectively). We normalized all variables and conducted multiple regression for three mechanism indices at each grid point of SCHL anomalies across the target region. And, we quantified the relative contributions of three mechanisms by multiplying the linear regression coefficients between the Nino3.4 index and each mechanism index by their multiple regression coefficients respectively (Eq. R3-1; We added it as Eq. 4 in the manuscript).

$$\frac{dChl}{dNino3.4}(\alpha) = \frac{\partial Chl}{\partial Runoff} \times \frac{dRunoff}{dNino3.4} + \frac{\partial Chl}{\partial TS-transport} \times \frac{dTS-transport}{dNino3.4} +$$

$$\frac{\partial Chl}{\partial Ekman-Upwelling} \times \frac{dEkman-Upwelling}{dNino3.4} + residual \qquad \text{(Eq. R3-1)}$$

The results indicate that both the Runoff- and the TS-transport-driven mechanism have the most significant contributions to the intensity of SCHL blooming in the target region. Although the Ekman Upwelling-driven mechanism exhibits a lower contribution compared to the others, it still exerts a meaningful influence on about half

of the other mechanisms. Spatially, the Runoff-driven mechanism primarily affects the central area of the target region (Figure R3-7a), while the TS-transport-driven mechanism plays a major role along a southwest-northeast diagonally (Figure R3-7b). In contrast, the Ekman Upwelling-driven mechanism uniformly influences the overall target region (Figure R3-7c). Especially, the runoff- and the TS-transport-driven mechanism contribute similarly to the SCHL blooming intensity in the target region. Although the Ekman upwelling-driven mechanism accounts for approximately 40 to 47% of the others, it still exerts a significant influence (Table R3-1). Consequently, we have replaced the paragraph that calculated the relative contribution for each mechanism from the joint-composite analyses previously we used to the regression method as follows:

(L313-349): So far, we have conducted a comprehensive evaluation and analysis of the complex mechanisms driving phytoplankton blooms in the ECS during the summers following the decaying phase of El Niño events. In addition to the runoff- and TS-transport-driven mechanisms suggested in previous studies, we introduced the Ekman upwelling-driven blooming mechanism. Subsequently, we quantified the relative contributions of these three mechanisms based on regression methods using (Eq. 4) as follows:

$$\frac{dChl}{dNino3.4}(\alpha) = \frac{\partial Chl}{\partial Runoff} \times \frac{dRunoff}{dNino3.4} + \frac{\partial Chl}{\partial TS-transport} \times \frac{dTS-transport}{dNino3.4} +$$

$$\frac{\partial Chl}{\partial Ekman-Upwelling} \times \frac{dEkman-Upwelling}{dNino3.4} + residual \qquad (4)$$

Firstly, we normalized all variables and conducted the multiple regression with respect to SCHL index in the target region as the dependent variable and three mechanism indices as independent variables. Secondly, we calculated each term by multiplying the linear regreesion coefficient between Nino3.4 index and each mechanism index by the corresponding multiple regression coefficient for each mechanism index. The combined effects of the three mechanisms and the residual term collectively account for the variation of the SCHL index in

the ECS region with respect to the El Niño events. Therefore, each term indicates how changes in three mechanism induced by ENSO cycle (as indicated by the Nino3.4 index) affect on phytoplankton blooms in the ECS region. The results show the effect of runoff-driven mechanism and the TS-transport mechanisms to SCHL blooming intensity in the target region are comparable (Table 1). In the case of the Ekman upwelling mechanism, it accounts for about 40% to 47% of contributions comapred to the other two mechanisms, yet still exerts a meaningful influence as well.

**Table R3-1.** Relative contributions of three mechanisms to SCHL blooming in the target region.

|  | $\alpha$ | Runoff | TS-transport | Ekman Upwelling |
|---|---|---|---|---|
| Contribution | 0.511 | 0.152 | 0.138 | 0.065 |

Additionally, we identified their spatial contributions to further elucidate the effects of each mechanism within the target region (Figure 9). We conducted multiple regression for three mechanism indices at each grid point of SCHL anomalies across the target region. Following the same way examined above, we calculated the spatial contributions by multiplying the linear regression coeffients of each mechanism with repsect to the Nino3.4 index. The results showed that the runoff-driven mechanism exhibited the strongest contribution at the center of the target region, driven by direct riverine nutrient inflow (though the model doesn't simulate P input) and nutrient supply through buoyancy upwelling induced by river discharge (Figure 9a). In contrast, the TS-transport-driven mechanism significantly contributes along a southwest-northeast diagonal direction, aligning with nutrient transport via the TS from the SCS (Figure 9b). Additionally, the newly proposed Ekman upwelling-driven mechanism, while relatively lower in overall contribution, shows significant and uniformly distributed impacts across the target region (Figure 9c). These results suggest that all mechanisms can affect comprehensively to induce phytoplankton blooms in the ECS during the following summers of the decaying phase of El Niño events. Therefore, considering all mechanisms is essential for accurately predicting the intensity of phytoplankton blooms.

[Figure]

**Figure R3-7.** Relative contributions of three mechanisms to SCHL blooming in the target region during summers following the decaying phase of El Niño events **(a)** Runoff-driven mechanism **(b)** TS-transport-driven mechanism **(c)** Ekman upwelling-driven mechanism.

• In describing the TOPAZ model, the authors stated Dunne et al. (2010, 2013) TOPAZ model is implemented in this study. As TOPAZ is developed in completely different context. I am not sure if the authors have done anything to tune the model parameters for ECS regions? If so, the model parameters should be listed in a table, if not in the main texts, should be in supplementary materials.

**Response:** We conducted the experiments using the GFDL-CM2.1 Earth System Model coupled with the TOPAZ biogeochemical model. As we coupled our in-house model to an already well-optimized marine biogeochemical model, we did not make any adjustments or tuning to the model parameters.

• English is mostly well written, but therse are places where comprehension is compromised due to incorrect English.

**Response:** Thank you for your comments on the English writing. We tried to improve the English description and expressions throughout the entire manuscript.

**Detailed comments:**

• Line 39: citation of Racault et al. (2017) is not appropriate, as that paper does not address ECS specifically.

**Response:** We have corrected with more appropriate references.

• Line 42: What does YECS stand for?
**Response:** Sorry for omitting the abbreviation. We have revised it in the manuscript as follows:

(L44) "Yellow and East China Seas (YECS)"

• Line 89: Fe is a "micronutrients", but others are macronutrients.
**Response:** We have revised it in the manuscript for clarify as follows:

(L102) "macronutrients & micronutrients"

• Figure 1: How were the anomalies calculated? Are they against annual mean? or a climatology of certain period? Need to be precisely stated.

**Response:** We have added the description of the anomaly as follows:

(L104-105) "All anomalies are calculated by removing the seasonal cycles and the linear trend."

• Line 102: Yamaguchi et al. (Year?)

**Response:** Corrected.

(L117) "Yamaguchi et al (2012)"

• Line 113-114: "This anomalous phytoplankton bloom during the boreal summer season of the following El Niño events has been explained by enhanced precipitation." This sentence is hard to read. Are you trying to say the anomalous phytoplankton bloom following the El Nino events has been explained by enhanced precipitation?

**Response:** In order to address the aims of our study and improve readability, we have revised the manuscript as follows:

(L132-135) "The anomalous phytoplankton blooms during the summers following El Niño events can be attributed to increased riverine discharge from the YR, driven by enhanced precipitation, and the substantial nutrient supply through the TS from the South China Sea (SCS)."

• Figure 2(e, and f): What do red and red squares stand for? Describe it in the legend.

**Response:** The red and blue squares represent the YRE and Extended YRE regions, respectively, and we have added the description to the Figure 2 caption.

• Figure 4a: What are the gray dots for?

**Response:** Figure 4a shows the relationship between the Nino3.4 DJF index and

area-averaged surface chlorophyll-a anomalies over the target region in GFDL-CM2.1 ESM used in the research. Gray dots represent the non-El Niño cases. We have added the description of the gray dots in Figure 4 caption.

• Line 173-174: Where is the observed patterns of NO3 and PO4?

**Response:** Annual nutrient distribution from World Ocean Atlas (WOA) reanalysis data reveals a $NO_3$-dominated pattern centered around the YRE, which is attributable to strong river discharge. In contrast, $PO_4$ concentrations remain relatively low throughout the entire ECS region.

[Figure]

**Figure R3-8.** Annual mean nutrient distribution **(a)** NO3 **(b)** PO4 concentrations

• Fig. 5: How did you simulate runoff? It appears there is no grid close to the coast due to coarse resolution.

**Response:** The model simulates the runoff in the grids where dots are displayed in Figure 5d-f.

• Line 260: " within the red box in Fig. 7g". The red box is in Fig. 7c.

**Response:** Corrected.

• Line 261-263: How deep could be the upwelling, and what is the nutrient

concentration of the upwelling horizon?

**Response:** The ECS is featured by shallow waters, with a maximum depth of approximately 50 meters, making it highly conducive to the upward transport of nutrients from the subsurface layer through upwelling.

Jing et al (2009) mentioned that the local winds in the northern coastal region of the South China Sea can induce upwelling within 100m depth of water column. This suggests that such upwelling is sufficient to transport nutrients to the surface layer, promoting the growth of phytoplankton and enhancing primary productivity in the ECS region.

• Line 274: "Joint composition analysis" may not be familiar to large number of readers, including myself. A reference should be provided.

**Response:** In the revised manuscript, we assessed the relative contributions of three mechanisms using a regression method instead of the joint-composite analyses.

Therefore, we have replaced the paragraph that calculated the relative contribution for each mechanism from the joint-composite analyses to the regression method.

• Fig 9(a) and Fig 7(a-c): The position of modelled and observed position of GPH anomalies are quite different in latitude. Need some explanations.

**Response:** Figures 7a–c show the location of the WNPAC in the model results. During the summer of the El Niño decaying phase, the WNPAC location in the observations (16.5°N–26.5°N / 130°E–155°E) and the model (13.5°N–26.5°N / 124.5°E–160.5°E) are slightly different in terms of zonal location.

We have added a description of the location of the WNPAC index in the model results shown in Figure 8c as the red box in the revised manuscript:

(L307-309) "The correlation between WSCL and the WNPAC index (13.5°N–26.5°N / 124.5°E–160.5°E; which is slightly broader than observations) calculated from GPH anomalies within the red box in Fig. 8c exhibits a significantly positive relationship (r = 0.45**) at the 99% confidence level"

• Line 368: What do you mean by "two seasons"?
**Response:** It refers to the El Niño peak phase (D(0)JF(1)), which represents two seasons prior to the El Niño decaying summers (JJA(1))

• Lines 370-371: Not clear about the correlation between what? Do you mean the correlation between observation and ESM results?

**Response:** Figures 11a–d illustrate the lagged relationship between the WNPAC index and observed SCHL concentration for the D(0)JF(1)–JJA(1) in Figure 11a-b and MAM(1)–JJA(1) in Figure 11c-d in observations. Figures 11e–h present the correlations between model results as same as Figure 11a-d. To prevent confusion between observations and models in Figure 11 (previously Figure 10 in the manuscript), we have added the additional explanation as follows:

(L421-422) "We found a significantly positive lagged relationship between the WNPAC index and blooming magnitude in the ECS region in both observations and ESM results (Figure 11a-d; Observations, Figure 11e-h; Model)."

---

## Referee Report (RR1)

**Review of Lee et al., Two different phytoplankton blooming mechanisms over the East China Sea during El-Niño decaying summers: Round 2**

March 5, 2025

**General comments**

- I appreciate the authors for thorough revision that clarified some of the questions I raised in the previous round of review, which allows me better understand the manuscript.

- When looked at the surface Chl a anomaly (SCHL), the anomaly due to El Nino events are really small (-0.06 to +0.06 mg Chl m$^{-3}$) (Figure 4). From observational measurement perspective, this magnitude of change is within the error of measurement. This is about 1/10 to 1/5 of the observed variation (Figure 10).

- Regarding the buoyancy upwelling mentioned as a mechanism for enhancing phytoplankton bloom (line 250 in the track changed document), I don't understand how that happens. I assume river runoff carries water of lower density, and the deep water upwells only when it becomes less dense than surface. I don't see how that is possible in this region. Estuary circulation may bring the subsurface water up to the surface, but it is only possible in the upstream of the estuary. I have to wonder where this may happen, and whether the location of upwelling due to estuary circulation is covered in the model domain. This mechanism is barely speculation without model data to support.

- The equation 4 is hard to follow in the context of multiple regression. The TOPAZ model resolves the $NO_3$, and $PO_4$ limitation, and phytoplankton growths are limited by the minimum of the two nutrient limitations. This equations suggests $PO_4$ limitation is ignored. If the purpose is to understand the contribution of different variables to the Chl anomaly ($\delta$Chl), then the equation should be in the form of:

$$\delta Chl = Intercept + a\,\delta Runoff + b\,\delta TStransport + c\,\delta EkmanUpwelling$$

  where $a$, $b$, and $c$ are regression coefficients, which tell the importance of the variable if variables are normalized as the authors indicated. Then comes another problem. As Runoff, TS-transportations, and Ekman-upwelling are all correlated with Chl anomaly, it suggests that those independent variables in the regression are correlated with each other, that is they are not independent.

- In conclusion, I am not convinced this work provides valuable insights on the mechanism of how El Nino events affect the bloom dynamics. I can't recommend for publications before those issues are resolved.

---

## Author Response (AR2)

**Review of "Two different phytoplankton blooming mechanisms over the East China Sea during El-Niño decaying summers" by Lee et al.**

**Comment on egusphere-2024-3406 | Anonymous Referee #1:**

The revision is moderately responsive to my comments. Ultimately, it's up to the authors to decide how they are to respond to the reviews. If a suggested change is not justified, the authors can choose not to make the change but they should not report in Reply changes they have not made. THEIR REPLY SHOULD BE CONSISTENT WITH THE REVISED MANUSCRIPT. PLEASE DOUBLE CHECK. Such inconsistencies add extra work on the reviewers. Such inconsistencies add extra work on the reviewers.

In the title and elsewhere, I suggest changing "El-Niño decaying summers" to "post-El Niño summers." On average, El Niño has decayed by an post El-Niño summer.

**Response:** We thank referee #1 for constructive comments and valuable time reviewing our manuscript.

We have replaced "El-Niño decaying summers" with "post-El Niño summers" in the entire of the manuscript, and changed the title to "Phytoplankton blooming mechanisms over the East China Sea during post-El Niño summers".

**Comment on egusphere-2024-3406 | Anonymous Referee #3:**

General comments

• I appreciate the authors for thorough revision that clarified some of the questions I raised in the previous round of review, which allows me better understand the manuscript.

**Response:** We indeed thank to the referee's constructive and valuable comments and suggestions to improving our manuscript.

• When looked at the surface Chl a anomaly (SCHL), the anomaly due to El Nino events are really small (-0.06 to +0.06 mg Chl m−3 ) (Figure 4). From observational measurement perspective, this magnitude of change is within the error of measurement. This is about 1/10 to 1/5 of the observed variation (Figure 10).

**Response:** We acknowledge a significant discrepancy in variability between satellite-derived surface chlorophyll-a (SCHL) data and the model outputs. In our study, we used SCHL data measured from ocean color satellite observations, which have been widely expected to be useful for detecting and analyzing the spatio-temporal distribution of SCHL from many previous studies (Kim et al., 2009; Yamada et al., 2004; Zhang et al., 2017). It is well established that satellite-based SCHL estimates tend to be substantially overestimated in coastal regions due to the turbidity from the large amounts of re-suspended bottom sediments (Kiyomoto et al., 2001; Siswanto et al., 2011; Yamaguchi et al., 2012). Additionally, Gong (2004) noted that high levels of colored dissolved organic matter (CDOM) from the Yangtze River can also contribute to overestimated SCHL concentrations.

Therefore, even though the model simulates about 20% of the observed range, it effectively captures the observed SCHL variability in response to climate variability. Notably, during the post-El Niño summer season, the correlation between the Nino3.4 index and the SCHL anomalies in the model results over the East China Sea (ECS) is

highly significant at the 99% confidence level, similar to the observations. Moreover, the lagged relationships (with a delay of one to two seasons; Figure 11 in the main text) are also highly significant with correlations above 0.5 for both observations and models, indicating that the model simulates the observed SCHL variability well.

• Regarding the buoyancy upwelling mentioned as a mechanism for enhancing phytoplankton bloom (line 250 in the track changed document), I don't understand how that happens. I assume river runoff carries water of lower density, and the deep water upwells only when it becomes less dense than surface. I don't see how that is possible in this region. Estuary circulation may bring the subsurface water up to the surface, but it is only possible in the upstream of the estuary. I have to wonder where this may happen, and whether the location of upwelling due to estuary circulation is covered in the model domain. This mechanism is barely speculation without model data to support.

**Response:** Generally, the presence of less dense water in the top layer of a water column typically creates a strong vertical density gradient between the surface and the subsurface layer, which greatly constrains water exchange. Chen (2008) refers to this as the "capping effect". However, in the Yangtze River estuary (YRE) over the ECS, the upper layer is characterized by the discharge of a large volume of low-salinity water into the coastal ocean. This freshwater plume spreads horizontally, creating a distinct boundary with the denser, high-salinity surrounding water, leading to a strong horizontal density gradient along the plume's edge. This gradient generates a pressure difference that drives a compensatory circulation to maintain water balance (see Figure 2 in Chen, 2008). Consequently, this circulation facilitates the entrainment of nutrient-rich subsurface water (Hill, 1998; Chen et al., 2003; Chen, 2000), as the pressure gradient forces denser, nutrient-rich water to upwell along the plume boundary, delivering nutrients into the euphotic zone where they fuel phytoplankton growth.

Indeed, when a strong flood occurs, despite the presence of low-salinity surface water, saltier subsurface water is observed on the ECS shelves (Hu et al.,

2001; Delcroix and Murtugudde, 2002). Chen (2008) mentioned that such observation supports the occurrence of water entrainment along coastal shelves driven by the enhanced riverine discharge. In addition, the TOPAZ model does not account for phosphorus (P) input via runoff; however, as shown in Figure 9 in the main text, the runoff mechanism has the most pronounced effect near the YRE, where phosphate ($PO_4$) limitation is dominant. This suggests that the dominant signal attributed to the runoff mechanism is primarily influenced by buoyancy-driven upwelling, which entrains $PO_4$-rich subsurface water into the surface layer. Lastly, we have revised the description of buoyancy-driven upwelling due to enhanced river discharge in more detailed in the main text to improve clarify for readers.

(L141-147): As the freshwater plume outflow surpasses the incoming discharge, a pronounced horizontal density gradient develops along the plume boundary. This gradient creates a pressure difference, which in turn drives a compensatory circulation to maintain the water balance. Essentially, the pressure gradient forces the denser, nutrient-rich water from below to rise along the plume boundary, thereby upwelling into the surface layer. Consequently, the nutrient-rich Kuroshio subsurface waters ascend along the ECS shelf edge, providing essential nutrients. This buoyancy-driven upwelling occurs independently of wind conditions, driven primarily by the physical properties of the subsurface waters in response to the enhanced river discharge.

• The equation 4 is hard to follow in the context of multiple regression. The TOPAZ model resolves the NO3, and PO4 limitation, and phytoplankton growths are limited by the minimum of the two nutrient limitations. This equations suggests PO4 limitation is ignored. If the purpose is to understand the contribution of different variables to the Chl anomaly (δChl), then the equation should be in the form of: δChl = Intercept+a δRunoff+b δT Stransport+c δEkmanUpwelling where a, b, and c are regression coefficients, which tell the importance of the variable if variables are normalized as the authors indicated. Then comes another problem. As Runoff, TS-transportations, and Ekmanupwelling are all correlated with Chl anomaly, it suggests that those independent variables in the regression are correlated with each other, that is they are not independent.

**Response:** We appreciate the referee's comments and suggestions regarding equation 4. The equation is intended to quantitatively decompose the influence of El Niño on SCHL anomalies (δChl) through three physical mechanisms—Runoff, TS-transport, and Ekman upwelling—by examining their independent sensitivities to changes in El Niño intensity (as indicated by Nino3.4 index). The equation is formulated as follows:

$$\frac{dChl}{dNino3.4}(\alpha) = \frac{\partial Chl}{\partial Runoff} \times \frac{dRunoff}{dNino3.4} + \frac{\partial Chl}{\partial TS-transport} \times \frac{dTS-transport}{dNino3.4} +$$

$$\frac{\partial Chl}{\partial Ekman-Upwelling} \times \frac{dEkman-Upwelling}{dNino3.4} + residual$$

In this equation, each partial derivative term, ∂Chl/∂X (where X represents the three blooming mechanisms: Runoff, TS-transport, Ekman-Upwelling) were calculated from the multiple regression analysis, which account for their co-relationship.

We acknowledge the referee's concern regarding potential multicollinearity among the independent variables (Runoff, TS-transport, Ekman Upwelling). To address this, we assessed multicollinearity using the Variance Inflation Factor (VIF), yielding the following values:

- Runoff mechanism: 1.265
- TS-transport mechanism: 1.08
- Ekman Upwelling: 1.214

These VIF values, ranging from 1.08 to 1.265, indicate minimal multicollinearity—generally, VIF values below 3 suggest negligible multicollinearity, while values above 5 may indicate significant concerns (Kock and Lynn, 2012; Kim, 2019). Given that all our values are well below this threshold, we conclude that the three mechanisms can be considered statistically independent enough to allow for reliable quantification of their individual contributions to SCHL variability.

In summary, our equation explicitly decomposes the influence of El Niño on SCHL anomalies via distinct physical mechanisms. By employing multiple regression coefficients (∂Chl/∂X), this approach effectively quantifies each mechanism's independent contribution, enabling a robust interpretation of their relative importance

and spatial contributions. Thus, we believe this method captures both the quantitative and spatial contributions of these mechanisms' impacts on SCHL anomalies.

**Reference**

Chen, C. A.: Buoyancy leads to high productivity of the Changjiang diluted water: a note, Acta Oceanol. Sin., 27, 133–140, 2008.

Chen, C. T. A.: The Three Gorges Dam: Reducing the upwelling and thus productivity in the East China Sea, Geophys. Res. Lett., 27, 381–383, https://doi.org/10.1029/1999GL002373, 2000.

Chen, C. T. A., Liu, C. T., Chuang, W. S., Yang, Y. J., Shiah, F. K., Tang, T. Y., and Chung, S. W.: Enhanced buoyancy and hence upwelling of subsurface Kuroshio waters after a typhoon in the southern East China Sea, J. Mar. Syst., 42, 65–79, https://doi.org/10.1016/S0924-7963(03)00065-4, 2003.

Delcroix, T. and Murtugudde, R.: Sea surface salinity changes in the East China Sea during 1997-2001: Influence of the Yangtze River, J. Geophys. Res. Ocean., 107, 1–11, https://doi.org/10.1029/2001jc000893, 2002.

DX Hu, WY Han, S. Z.: Land-Ocean Interaction in Changjiang and Zhujiang Estuaries and Adjacent Sea Areas, China Ocean Press, 2001.

Gong, G. C.: Absorption coefficients of colored dissolved organic matter in the surface waters of the East China Sea, Terr. Atmos. Ocean. Sci., 15, 75–87, https://doi.org/10.3319/tao.2004.15.1.75(o), 2004.

Hill, A. E.: Buoyancy effects in coastal and shelf seas, Sea, 21–62, 1998.

Kim: Statistical Results, Korean J. Anesthesiol., 72, 558–569, 2019.

Kim, H. C., Yamaguchi, H., Yoo, S., Zhu, J., Okamura, K., Kiyomoto, Y., Tanaka, K., Kim, S. W., Park, T., Oh, I. S., and Ishizaka, J.: Distribution of Changjiang Diluted Water detected by satellite chlorophyll-a and its interannual variation during 1998-2007, J. Oceanogr., 65, 129–135, https://doi.org/10.1007/s10872-009-0013-0, 2009.

Kiyomoto, Y., Iseki, K., and Okamura, K.: Ocean color satellite imagery and shipboard measurements of cholorophyll a and suspended particulate matter distributionin the East China Sea, J. Oceanogr., 57, 37–45, https://doi.org/10.1023/A:1011170619482, 2001.

Kock, N. and Lynn, G. S.: Lateral collinearity and misleading results in variancebased SEM: An illustration and recommendations, J. Assoc. Inf. Syst., 13, 546–580, https://doi.org/10.17705/1jais.00302, 2012.

Siswanto, E., Tang, J., Yamaguchi, H., Ahn, Y. H., Ishizaka, J., Yoo, S., Kim, S. W., Kiyomoto, Y., Yamada, K., Chiang, C., and Kawamura, H.: Empirical ocean-color algorithms to retrieve chlorophyll-a, total suspended matter, and colored dissolved organic matter absorption coefficient in the Yellow and East China Seas, J. Oceanogr., 67, 627–650, https://doi.org/10.1007/s10872-011-0062-z, 2011.

Yamada, K., Ishizaka, J., Yoo, S., Kim, H. C., and Chiba, S.: Seasonal and interannual variability of sea surface chlorophyll a concentration in the Japan/East Sea (JES), Prog. Oceanogr., 61, 193–211, https://doi.org/10.1016/j.pocean.2004.06.001, 2004.

Yamaguchi, H., Kim, H. C., Son, Y. B., Kim, S. W., Okamura, K., Kiyomoto, Y., and Ishizaka, J.: Seasonal and summer interannual variations of SeaWiFS chlorophyll a in the Yellow Sea and East China Sea, Prog. Oceanogr., 105, 22–29, https://doi.org/10.1016/j.pocean.2012.04.004, 2012.

Zhang, H., Qiu, Z., Sun, D., Wang, S., and He, Y.: Seasonal and interannual variability of satellite-derived chlorophyll-a (2000-2012) in the Bohai Sea, China, Remote Sens., 9, https://doi.org/10.3390/rs9060582, 2017.

---

## Editor Decision (ED2)

**Review of Lee et al., Two different phytoplankton blooming mechanisms over the East China Sea during El-Niño decaying summers: Round 2**

March 18, 2025

**General comments**

- The argument that satellite remote sensing Chla overestimate real Chl does not hold. Algorithms for validating chl a from ocean color in coastal ocean have been well developed. For example, Figure 2 of Zhang et al (2017), cited in your last response, suggests the measured Chl and remote sensing Chl are very consistent at large range of Chl a concentrations in ECS. I have to wonder if the low Chl a anomaly simulated in your study is due to some problems with the biogeochemistry model that haven't been tuned for ECS regions, or other problems due to low resolution. Modelling work by Chen et al. (2021) shows good agreements between model and remote sensing Chl a (Fig. 2 of Chen et al., 2021), with chl anomaly around 1 mg m$^{-3}$ by visual estimation. Wu et al. (2023. Fig. 7) show that the difference in Chl a due to changes in river discharge in different phases of ENSO is between -1 and 1 mg m$^{-3}$. I am not sure how your modeled river discharge of water and nutrients are consistent with ground truth. It is important to have a solid discuss why the Chl anomaly is so small.

- Regarding the buoyancy-driven upwelling driven by river water plume is beyond my knowledge limit. I would appreciate any reviewers with

strong physics background to make the judge. However, As the buoyancy driven upwelling is argued to be the vector of runoff driven Chl a anomaly, but not quantified. It is only a hypothesis, and needs to be discussed, along with direct nutrient input from river water. Relevant literature that may collaborate the hypothesis should be cited.

- Regarding Equation 4, I appreciate the VIF analysis, which is robust. This should be added to the presentation of results. However, the expression of the equation 4 does not agree with your text. Following your description, I guess $\frac{\delta Chl}{\delta Runoff}$ is the partial coefficient of Runoff on Chl change in the multiple regression between Chl a and three mechanisms. Then, is $\frac{\delta Runoff}{\delta NO_3}$ the regression coefficient between ENSO index and Runoff? If that is correct, then my question is how you deal with the effects of runoff on $PO_4$? That maybe ok for the effects of runoff on nutrient supply, as there is no $PO_4$ in runoff. But how do you quantify the impact of upwelling (Ekman or buoyancy) and TS transport on $PO_4$, as either $NO_3$ or $PO_4$ may be limiting phytoplankton growth in your model. This needs to be clearly and rigorously explained in the equations and texts.

**References**

1. CHEN D., LIU Q., and YIN K., 2021. Numerical Study of the Three Gorges Dam Influences on Chlorophyll-a in the Changjiang Estuary and the Adjacent East China Sea. J. Ocean Univ. China (Oceanic and Coastal Sea Research). https://doi.org/10.1007/s11802-021-4430-z

2. Wu, Q.; Wang, X.; He, Y.; Zheng, J. The Relationship between Chlorophyll Concentration and ENSO Events and Possible Mechanisms off the Changjiang River Estuary. Remote Sens. 2023, 15, 2384. https://doi.org/10.3390/rs15092384

3. Zhang, H., Qiu, Z., Sun, D., Wang, S., and He, Y.: Seasonal and interannual variability of satellite-derived chlorophyll-a (2000-2012) in the Bohai Sea, China. Remote Sens., 9, https://doi.org/10.3390/rs9060582, 2017.

---

## Author Response (AR3)

**Review of "Two different phytoplankton blooming mechanisms over the East China Sea during El-Niño decaying summers" by Lee et al.**

**Comment on egusphere-2024-3406 | Editor Comment:**

In addition to the reviewer's comments, I also noticed that "chlorophyll a anomaly" in all relevant figures is currently labeled as "chlorophyll a". Please change it to "chlorophyll a anomaly" or an appropriate symbol.

**Response:** We sincerely appreciate the editor's valuable time and support in evaluating our work.

Following the editor's suggestion, we have updated the labels of all the relevant figures to surface chlorophyll-a (SCHL) in the main text and supplementary figures to "SCHL anomaly" for consistency.

**Comment on egusphere-2024-3406 | Anonymous Referee #3:**

General comments

- The argument that satellite remote sensing Chla overestimate real Chl does not hold.

Algorithms for validating chl a from ocean color in coastal ocean have been well developed. For example, Figure 2 of Zhang et al (2017), cited in your last response, suggests the measured Chl and remote sensing Chl are very consistent at large range of Chl a concentrations in ECS. I have to wonder if the low Chl a anomaly simulated in your study is due to some problems with the biogeochemistry model that haven't been tuned for ECS regions, or other problems due to low resolution. Modelling work by Chen et al. (2021) shows good agreements between model and remote sensing Chl a (Fig. 2 of Chen et al., 2021), with chl anomaly around 1 mg m−3 by visual estimation. Wu et al. (2023. Fig. 7) show that the difference in Chl a due to changes in river discharge in different phases of ENSO is between -1 and 1 mg m−3. I am not sure how your modeled river discharge of water and nutrients are consistent with ground truth. It is important to have a solid discuss why the Chl anomaly is so small.

**Response:** Actually, our model is a global-scale climate model with a relatively coarse resolution (1°×1°). In contrast, the model used in the reference suggested by the referee (Chen et al., 2021) is a high-resolution regional model simulation (ROMS) concentrated on the East Asian marginal seas, including the East China Sea (ECS). Therefore, the ROMS model—due to its higher spatial resolution—is better able to simulate observed surface chlorophyll-a (SCHL) concentrations, especially in narrow coastal regions like the Yangtze River estuary, compared to our coarser-resolution model.

Additionally, we compared modeled river discharge and observed to address the referee's concern. Since the model does not provide the river discharge variable from land, we indirectly evaluated the Yangtze River discharge by comparing the modeled liquid runoff over a spread grid domain (26.5°N–32.5°N / 121.5°E–125.5°E; approximately 660 km × 440 km) with observation-based discharge estimates. According to Guo et al (2018), the average annual discharge of the Yangtze River is

reported as 10,870, 13,620, and 28,400 m³/s at Cuntan, Yichang, and Datong stations, respectively. To represent discharge near the estuary, we used the value from the Datong station (28,400 m³/s) for comparison. The corresponding area-averaged runoff from observations, when distributed uniformly across the estuary region, is approximately $9.78 \times 10^{-5}$ kg/m²/s. In contrast, the modeled mean runoff over the same region is $7.25 \times 10^{-5}$ kg/m²/s, which simulates rather well, about 74.1% of the observation-based estimate.

• Regarding the buoyancy-driven upwelling driven by river water plume is beyond my knowledge limit. I would appreciate any reviewers with 1 strong physics background to make the judge. However, As the buoyancy driven upwelling is argued to be the vector of runoff driven Chl a anomaly, but not quantified. It is only a hypothesis, and needs to be discussed, along with direct nutrient input from river water. Relevant literature that may collaborate the hypothesis should be cited.

**Response:** In response to comments from another referee, we have acknowledged the potential role of buoyancy-driven upwelling induced by river runoff in the Yangtze River Estuary (YRE). In the revised manuscript, we highlighted that, beyond the direct nutrient input from runoff, subsurface nutrient supply via buoyancy-driven upwelling could also play a significant role. The reference to Chen (2008), which explains this mechanism, was suggested by the referee. Additionally, we have incorporated several relevant studies—including Chen et al. (2003), Chen (2000), and Hill (1998)—that emphasize how strong river discharge can drive buoyancy-induced upwelling in coastal systems. These references have been cited to strengthen our discussion of this mechanism in the manuscript.

• Regarding Equation 4, I appreciate the VIF analysis, which is robust. This should be added to the presentation of results. However, the expression of the equation 4 does not agree with your text. Following your description, I guess δChl/δRunof f is the partial

coefficient of Runoff on Chl change in the multiple regression between Chl a and three mechanisms. Then, is δRunoff/δNO3 the regression coefficient between ENSO index and Runoff? If that is correct, then my question is how you deal with the effects of runoff on PO4? That maybe ok for the effects of runoff on nutrient supply, as there is no PO4 in runoff. But how do you quantify the impact of upwelling (Ekman or buoyancy) and TS transport on PO4, as either NO3 or PO4 may be limiting phytoplankton growth in your model. This needs to be clearly and rigorously explained in the equations and texts.

**Response:** In the previous review, we calculated the variance inflation factor (VIF) for each mechanism to assess multicollinearity among the independent variables. As the referee suggested, we have added VIF values for each mechanism in the main text and included a brief description of typical VIF criteria that indicate whether they are statistically independent enough to be reliably quantified.

(L334-339): Before quantitatively assessing the relative contributions of each mechanism, we evaluated potential multicollinearity among the three mechanisms by calculating the Variance Inflation Factor (VIF). The VIF values for the three mechanisms—Runoff (1.265), TS-transport (1.08), and Ekman Upwelling (1.214)—ranged from 1.08 to 1.265, indicating minimal multicollinearity. VIF values below 3 are typically considered negligible multicollinearity, suggesting that the three mechanisms are statistically independent (Kock and Lynn, 2012; Kim, 2019).

Following the referee's suggestion, we conducted a quantitative assessment of the contributions of each mechanism to nutrient variabilities using Equation 4 in the main text. While the equation structure remained unchanged, we applied it to nitrate ($NO_3$) and phosphate ($PO_4$) instead of surface chlorophyll-a (SCHL). For the $NO_3$, the TS-transport mechanism was evaluated in the same method as for $PO_4$, with $NO_3$ concentrations substituted into equation 3 in the main text.

$NO_3$ results showed that the runoff mechanism emerged as the dominant

contributor across the entire target region, with a contribution notably higher than those of the TS-transport and Ekman upwelling mechanisms (Table R1). Spatially, the strongest influence of the runoff mechanism was concentrated in the YRE, as expected, due to the model's explicit simulations of riverine $NO_3$ input. The TS-transport and Ekman upwelling mechanisms exhibited spatial patterns that were closely similar to those of SCHL (Fig. R1 and Fig. 9 in the main text).

Regarding $PO_4$, the runoff mechanism also showed the greatest influence overall—approximately twice that of the TS-transport mechanism (Table R1). However, the Ekman upwelling mechanism showed a negative effect on $PO_4$ concentrations across the entire target region but exhibited a localized positive influence in the YRE region, where phosphorus (P) limitation is known to prevail (Fig. R2 and Table R1). Notably, the impact of Ekman upwelling was most modest in the central-northern part of the target region. The runoff mechanism, showed its strongest effect on $PO_4$ concentrations slightly offshore from the YRE, indicating that even in the absence of direct riverine $PO_4$ input, buoyancy-driven upwelling may substantially contribute to nutrient enrichment. The spatial distribution of the TS-transport mechanism's effect on $PO_4$ was broadly consistent with those observed for both SCHL and $NO_3$.

**Table R1.** Relative contributions of three mechanisms to nutrients ($NO_3$ and $PO_4$).

|  | $\alpha$ | Runoff | TS-transport | Ekman Upwelling |
|---|---|---|---|---|
| Contribution to $NO_3$ | 0.32 | 0.177 | 0.076 | 0.033 |
| Contribution to $PO_4$ | 0.366 | 0.113 | 0.059 | -0.035 |
| Contribution to $PO_4$ (YRE region) | 0.218 | 0.122 | 0.088 | 0.027 |

[Figure]

**Figure R1**. Relative contributions of three mechanisms to $NO_3$ anomaly in the target region during summers following the decaying phase of El Niño events (a) Runoff-driven mechanism (b) TS-transport-driven mechanism (c) Ekman upwelling-driven mechanism.

[Figure]

**Figure R2**. Relative contributions of three mechanisms to $PO_4$ anomaly in the target region during summers following the decaying phase of El Niño events (a) Runoff-driven mechanism (b) TS-transport-driven mechanism (c) Ekman upwelling-driven mechanism.

**Reference**

Kim: Statistical Results, Korean J. Anesthesiol., 72, 558–569, 2019.

Kock, N. and Lynn, G. S.: Lateral collinearity and misleading results in variance-based SEM: An illustration and recommendations, J. Assoc. Inf. Syst., 13, 546–580, https://doi.org/10.17705/1jais.00302, 2012.

Guo, L., Su, N., Zhu, C., and He, Q.: How have the river discharges and sediment loads changed in the Changjiang River basin downstream of the Three Gorges Dam?, J. Hydrol., 560, 259–274, https://doi.org/10.1016/j.jhydrol.2018.03.035, 2018.

Kim: Statistical Results, Korean J. Anesthesiol., 72, 558–569, 2019.

Kock, N. and Lynn, G. S.: Lateral collinearity and misleading results in variance-based SEM: An illustration and recommendations, J. Assoc. Inf. Syst., 13, 546–580, https://doi.org/10.17705/1jais.00302, 2012.

Chen, C. A.: Buoyancy leads to high productivity of the Changjiang diluted water: a note, Acta Oceanol. Sin., 27, 133–140, 2008.

Guo, L., Su, N., Zhu, C., and He, Q.: How have the river discharges and sediment loads changed in the Changjiang River basin downstream of the Three Gorges Dam?, J. Hydrol., 560, 259–274, https://doi.org/10.1016/j.jhydrol.2018.03.035, 2018.

Kim: Statistical Results, Korean J. Anesthesiol., 72, 558–569, 2019.

Kock, N. and Lynn, G. S.: Lateral collinearity and misleading results in variance-based SEM: An illustration and recommendations, J. Assoc. Inf. Syst., 13, 546–580, https://doi.org/10.17705/1jais.00302, 2012.

Chen, C. A.: Buoyancy leads to high productivity of the Changjiang diluted water: a note, Acta Oceanol. Sin., 27, 133–140, 2008.

Chen, C. T. A.: The Three Gorges Dam: Reducing the upwelling and thus productivity in the East China Sea, Geophys. Res. Lett., 27, 381–383, https://doi.org/10.1029/1999GL002373, 2000.

Chen, C. T. A., Liu, C. T., Chuang, W. S., Yang, Y. J., Shiah, F. K., Tang, T. Y., and Chung, S. W.: Enhanced buoyancy and hence upwelling of subsurface Kuroshio waters after a typhoon in the southern East China Sea, J. Mar. Syst., 42, 65–79, https://doi.org/10.1016/S0924-7963(03)00065-4, 2003.

Guo, L., Su, N., Zhu, C., and He, Q.: How have the river discharges and sediment loads changed in the Changjiang River basin downstream of the Three Gorges Dam?, J. Hydrol.,

560, 259–274, https://doi.org/10.1016/j.jhydrol.2018.03.035, 2018.

Hill, A. E.: Buoyancy effects in coastal and shelf seas, Sea, 21–62, 1998.

Kim: Statistical Results, Korean J. Anesthesiol., 72, 558–569, 2019.

Kock, N. and Lynn, G. S.: Lateral collinearity and misleading results in variance-based SEM: An illustration and recommendations, J. Assoc. Inf. Syst., 13, 546–580, https://doi.org/10.17705/1jais.00302, 2012.

---

## Author Response (AR4)

**Review of "Two different phytoplankton blooming mechanisms over the East China Sea during El-Niño decaying summers" by Lee et al.**

**Comment on egusphere-2024-3406 | Editor Comment:**

Two of the three reviewers (including the first round of review) were concerned with the coarse resolutions of the model used. You responded in detail but failed to include any relevant revisions in the manuscript. As readers may also raise similar concerns in the future, please, in the manuscript, briefly discuss the potential weaknesses of the model and explain why this model meets the requirements of your study.

**Response:** We are grateful to the editor for your time and support in reviewing our work.

In response to the editor's suggestion, we have added further explanation regarding the limitations of the model resolution, as well as the validity of the model-based results in the manuscript as follows:

(L447-454): In our study, we used a global climate model to investigate how large-scale climate variability influences oceanic biogeochemical processes. The model has a relatively coarse resolution of 1°×1° across the global domain, which limits its ability to resolve small-scale eddies and coastal upwelling, potentially leading to an underestimation of SCHL variability. Despite this limitation, the model reasonably captures the observed spatial and temporal patterns of SCHL variability, allowing us to effectively examine the physical mechanisms driving these variations. However, we acknowledge that the quantitative contribution of each physical process could be resolution-dependent. Therefore, future studies using higher-resolution models would be valuable for providing a more precise quantification of these processes.

L112-113: Change "surface chlorophyll-a anomalies (SCHL) to "surface chlorophyll-a (SCHL) anomalies".

**Response:** Corrected.

L164: Change "anomalous surface chlorophyll (SCHL) blooms" to "anomalous SCHL

blooms". SCHL has already been defined in L112-113.

**Response:** Corrected.